# Discrete Diffusion Models with MLLMs for Unified Medical Multimodal Generation

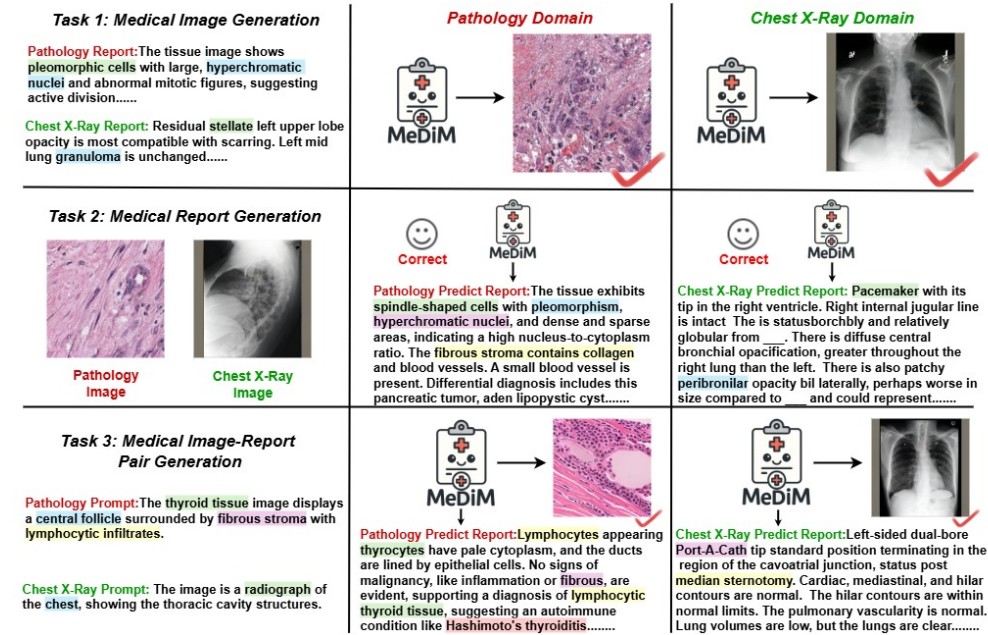

Figure 1: **MeDiM**, the *first medical discrete diffusion model*, is a flexible multimodal generator that simultaneously supports: **(i)** medical image generation from clinical reports, **(ii)** report generation from medical images, and **(iii)** joint synthesis of image–report pairs. Zoom in for a better view.

## Abstract

Recent advances in generative medical models are often constrained by modality-specific scenarios that hinder the integration of complementary evidence, such as imaging, pathology, and clinical notes. This fragmentation limits their development to true foundation models that empower medical AI agents to learn from and predict across the full spectrum of biomedical knowledge. To address these challenges, we propose **MeDiM**, the first medical discrete diffusion model that learns shared distributions across different medical modalities without requiring modality-specific components. MeDiM unifies multiple generative tasks: it flexibly translates between images and text or jointly produces image–report pairs across domains in response to user prompts. It builds on a discrete diffusion framework that unifies vision and language representations by modeling their shared probabilistic distribution. To empower the diffusion process to support unified and versatile medical generation, we employ a multimodal large language model (MLLM) as the diffusion backbone, leveraging its rich prior knowledge and cross-modal reasoning abilities. Because MLLMs are trained with causal (autoregressive) masking while diffusion denoising benefits from bidirectional context, MeDiM introduces two key designs: 1) *removing the causal attention mask* to enable a fully bidirectional information flow essential for mutual alignment, and 2) *injecting continuous timestep embeddings* to make the MLLM aware of the diffusion steps. Extensive experiments validate MeDiM as a unified foundation model capable of high-fidelity medical generation across various modalities, including

medical image generation (16.60 FID on MIMIC-CXR; 24.19 FID on PathGen) and report generation (0.2650 METEOR on MIMIC-CXR; 0.2580 METEOR on PathGen). In addition, the jointly generated medical image-report pairs improve the downstream task performance (+6.43% BLEU-1, +18.57% BLEU-2, +31.58% BLEU-3, and +4.80% METEOR in PathGen), enabling the use of multimodal inputs and the production of coherent, clinically grounded outputs..

# 1 INTRODUCTION

Modern medical systems and doctors rely on synthesizing multimodal evidence, encompassing radiology images, digital pathology images, EHR info and clinical reports. However, most existing medical AI models remain limited to isolated modalities (Moor et al., 2023). They often face limited insights when interpreting complex cases. For instance, current AI tools are unable to jointly analyze a lung nodule's imaging with its corresponding biopsy mutation status to predict treatment resistance, or generate clinically grounded images (e.g., counterfactual follow-up radiological scan or representative pathology patches) that visualize likely outcomes under different therapies. Bridging this gap demands a foundational shift: a unified, domain-aware multimodal system capable of understanding heterogeneous inputs, while generating clinically meaningful outputs. Such a system would directly address the challenge of cross-modal alignment in the medical context and serve as a foundation for medical AI agents that can learn from and generate across the full spectrum of biomedical knowledge.

Medical multimodal synthesis can represent a promising direction toward generalist medical AI agents, but existing methods remain limited. Medical-specific models like PairAug (Xie et al., 2024b) and MedM2G (Zhan et al., 2024) either rely on disconnected external models, preventing strict semantic alignment, or use modality-specific components that are difficult to adapt to multiply modality. In contrast, the natural image domain has witnessed the emergence of unified models (Team, 2024a; Xie et al., 2024a; Wu et al., 2024a; Yang et al., 2025) that, within a single framework, simultaneously support both generation and understanding tasks without the need for modality-specific designs. Liquid (Wu et al., 2024a) extends a pre-trained large language model (LLM) into a unified multimodal auto-regressive (AR) framework, allowing images and text to share a token space for both visual understanding and generation without altering the LLM architecture. Swerdlow et al. (2025) note that while AR models excel in text, their token-by-token prediction limits efficiency, motivating a unified multimodal discrete diffusion model that enables higher-quality, diverse, and controllable generation. Yang et al. (2025) further propose MMaDA, employing a unified diffusion architecture to jointly model image and text distributions. However, it does not support the paired generation of image-text outputs, a critical capability needed to address the medical challenges outlined above.

To our knowledge, no such unified models currently exist in the medical domain that could synthesize multimodal information while supporting multimodal generation (see Fig. 2). In this work, we propose **MeDiM**, the first medical discrete diffusion model that simultaneously models shared distributions across different modalities. Compared to domain-specific expert models, MeDiM can simultaneously perform diverse medical tasks across multiple medical modalities and domains, including medical image/report generation, and medical paired image–report synthesis (as shown in Fig. 1). A core design is the use of a Multimodal Large Language Model (MLLM) as the backbone for the diffusion process. Pre-trained MLLMs provide strong distribution-alignment priors from large-scale vision–language pretraining, making them powerful guides for multimodal generation. Their increasingly unified architectures are particularly well suited for paired image–report generation. Unlike MMaDA (Yang et al., 2025), which is restricted to diffusion-based MLLM backbones (Nie et al., 2025), our MeDiM can extend to a broader class of autoregressive (AR) MLLMs, offering greater generality and flexibility. However, adapting MLLMs to discrete diffusion introduces a fundamental mismatch: MLLMs are trained with a causal (autoregressive) attention mask, while the multimodal diffusion denoising process is inherently non-causal (e.g., requires bidirectional context). To resolve this, MeDiM incorporates two key modifications: (1) **causal attention removal**, enabling full bidirectional information flow for improved cross-modal alignment, and (2) **injecting continuous timestep embeddings**, allowing the MLLM to track diffusion steps. In addition, we integrate adaptive layer normalization (AdaLN) (Perez et al., 2018; Brock et al., 2018; Karras et al., 2019) to further stabilize the training and enhance its generative capability.

Figure 2: **Architectural comparison of medical multimodal models. ("BACKBONE")** indicates the backbone adopted in each framework. Prior approaches (A-D) cannot perform paired generation and suffer from other key limitations, such as requiring modality-specific components (A, B), inference inefficiency (C), or backbone inflexibility (D). In contrast, our model, MeDiM (E), provides a unified framework designed to overcome these challenges.

Our experiments demonstrate that MeDiM can function as a versatile foundation model for unifying various medical generative tasks: 1) For medical image generation, MeDiM achieved *state-of-the-art (SoTA) Frechet Inception Distances (FID)* on the MIMIC-CXR (Johnson et al., 2019) and PathGen (Sun et al., 2024b) datasets, respectively (Tab. 1 and Tab. 2; 16.60 FID and 24.19 FID), generating *high-fidelity* medical images across *different modalities* (Fig. 4; robust and high visual quality). 2) For medical report generation, MeDiM generated corresponding clinical reports from input medical images, demonstrating *semantic alignment* with target reports in MIMIC-CXR and PathGen datasets, respectively (Tab. 3; METEOR score of 0.265 and 0.258). 3) MeDiM generates *highly consistent*(Fig. 5a; higher consistency score in both large vision–language models (VLM) and human evaluation) medical image–report pairs. The generated medical image–report pairs can further *improve the performance of VLM* on downstream medical report generation tasks (Fig. 5c; +6.43% BLEU-1, +18.57% BLEU-2, +31.58% BLEU-3, and +4.80% METEOR in Path-Gen). 4) Comparative *analyses of backbone choices* (Sec. D; improve with MLLM backbone) reveal that MLLM backbones are particularly well-suited for multimodal paired generation in medical discrete diffusion models.

In summary, the key contributions of this work can be distinguished in the following aspects:

• We propose MeDiM, the first medical discrete diffusion model that models shared distributions in different medical modalities, without requiring modality-specific components.

• MLLMs with distribution-alignment priors are identified as superior backbones for discrete diffusion models in medical multimodal generation.

• MeDiM demonstrates state-of-the-art (SoTA) or competitive performance in unified medical image analysis and generation tasks, with the generated medical image–report pairs improving the performance of medical vision–language models (VLMs).

## 2 METHOD

We propose MeDiM, the first discrete diffusion model with MLLM designed for medical multimodal generation. The framework aims to jointly model the shared distributions between medical images and reports without requiring additional modality-specific components, thereby providing flexibility to accommodate more medical modalities. Fig. 3 illustrates the overall framework of MeDiM. This section introduces discrete diffusion models in the medical domain and MeDiM's architectural design. Related background can be found in Sec. A of appendices.

### 2.1 DISCRETE DIFFUSION MODELS

Diffusion models (Ho et al., 2020; Rombach et al., 2022) are probabilistic generative models that learn to approximate data distributions by sequentially corrupting and denoising samples. The forward diffusion process gradually perturbs the data sample $x_0$ with noise over a sequence of timesteps $t$, producing a latent distribution $q(x_t)$:

$$x_t \sim q(x_t \mid x_0) = \mathcal{N}\big(x_t; \sqrt{\bar{\alpha}_t}x_0, (1 - \bar{\alpha}_t)\mathbf{I}\big), \tag{1}$$

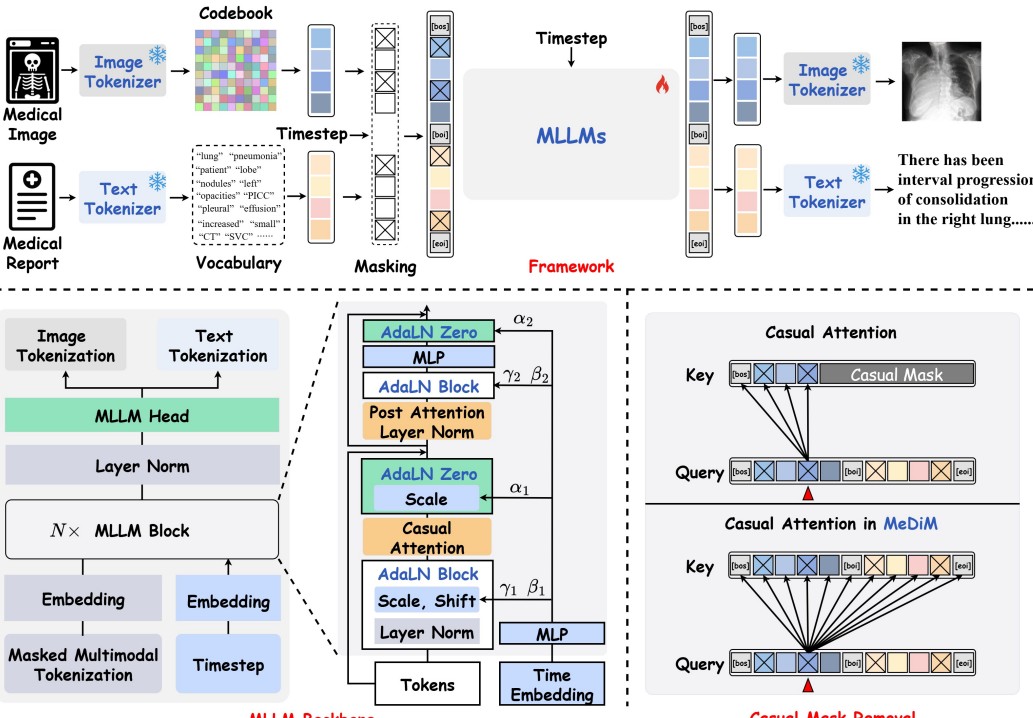

Figure 3: **Overview of the MeDiM architecture.** The framework integrates an MLLM backbone within a discrete diffusion process for unified medical multimodal generation. During the forward process, data is tokenized and diffused over timesteps. The MLLM is then trained to reverse this process. Key architectural adaptations, including causal attention removal, timestep embeddings, and AdaLN, adapt the autoregressive MLLM for the bidirectional denoising required for unified medical generation.

where $\bar{\alpha}_t = \prod_{s=1}^{t}(1 - \beta_s)$ denotes the cumulative retention coefficient and $\beta_t$ represents the pre-defined noise variance at timestep $t$. The reverse diffusion process involves learning a parameterized denoising model $\epsilon(.)$ to iteratively reconstruct the original data from noisy inputs:

$$x_{t-1} = \frac{1}{\sqrt{\alpha_t}}\left(x_t - \frac{1-\alpha_t}{\sqrt{1-\bar{\alpha}_t}}\,\epsilon(x_t, t)\right) + \sqrt{\frac{1-\bar{\alpha}_{t-1}}{1-\bar{\alpha}_t}\,\beta_t}\,z, \quad z \sim \mathcal{N}(0, I) \tag{2}$$

We introduce a discrete diffusion model (Sohl-Dickstein et al., 2015) for medical multimodal generation that jointly models medical images and reports within a shared probabilistic space.

**Forward Diffusion.** Our medical discrete diffusion models operate directly on sequences of discrete symbols $x_0$ consist of report tokenizations $x_{r0}$ and quantized medical image tokens $x_{i0}$ encoded by VQ-VAE (Van Den Oord et al., 2017). The forward diffusion process is formulated as a Markov chain $q(x_t \mid x_{t-1})$, where original symbols are gradually replaced with noise symbols, until the distribution converges to an approximate uniform distribution at a large timestep $T$. This process is parameterized by a transition matrix $Q_t \in \mathbb{R}^{K \times K}$, where $K$ denotes the sum of vocabulary size for the text tokenizer and VQ-VAE codebook. The elements of $Q_t$ represent the transition probabilities between discrete states, i.e., $[Q_t]_{ij}$ is the probability of transitioning from state $i$ at timestep $t-1$ to state $j$ at timestep $t$. Considering that absorbing transition matrices yield better performance in multimodal tasks (Austin et al., 2021; Lou et al., 2023), MeDiM introduces a special [MASK] token as an absorbing state, which serves as the noise symbol during the forward diffusion process. Please note that with the introduction of the additional [MASK] token, the dimension of the transition matrix is expanded to $Q_t \in \mathbb{R}^{(K+1) \times (K+1)}$. Accordingly, the transition matrix $Q_t$ for the absorbing state formulation can be expressed as:

$$Q_t = \alpha_t I + (1 - \alpha_t)\,\mathbf{1}\,e_m^\top, \tag{3}$$

here, $\alpha_t \in [0, 1]$ is the retention probability, $\mathbf{1}$ is a vector filled with ones, and $e_m$ indicates the canonical basis vector that activates only the absorbing [MASK] state $m$. This construction ensures

that once a token is replaced by the [MASK] symbol, it remains in that state for all subsequent timesteps. Consequently, the forward transition distribution is given by a categorical distribution:

$$x_0 = [x_{r0}, x_{i0}]$$
$$q(x_t \mid x_0) = \text{Cat}(x_t; p = \bar{Q}_t x_0), \tag{4}$$

where $\bar{Q}_t = \prod_{s=1}^{t} Q_s$. This implies that, as the number of timesteps increases, the input sequence is progressively replaced by [MASK] tokens, and at a sufficiently large timestep, the distribution converges to the absorbing state, providing a well-defined initialization for the reverse diffusion process.

**Reverse Diffusion.** The reverse diffusion process aims to reconstruct the original data sequence from noisy inputs by progressively recovering masked or transitioned symbols. Specifically, given a corrupted sequence $x_t$ at timestep $t$, the discrete diffusion model parameterizes the reverse transition distribution $p_\theta(x_{t-1} \mid x_t)$, which estimates the probability of recovering the clean symbol at the previous step. Formally, this process is defined as a categorical distribution over the shared vocabulary space:

$$p_\theta(x_{t-1} \mid x_t) = \text{Cat}(x_{t-1}; \epsilon(x_t, t)), \tag{5}$$

where $\epsilon(x_t, t) \in \Delta^{K+1}$ denotes the predicted categorical probabilities at timestep $t$, and $\Delta^{K+1}$ is the probability simplex over the extended vocabulary including the [MASK] token.

To effectively model multimodal medical data, we parameterize $\epsilon(x_t, t)$ using a backbone network $f_\theta(\cdot)$ built upon an MLLM. MeDiM incorporates timestep embeddings into the MLLM backbone, ensuring temporal conditioning across the transition trajectory. At each step, $f_\theta$ leverages prior alignment between medical image tokens and report tokens. Consequently, the reverse chain iteratively recovers $x_0 = [x_{r0}, x_{i0}]$ from a fully masked initialization, producing coherent paired outputs that align medical visual and textual modalities. The training objective (Sahoo et al., 2024) can be defined as the expected negative log-likelihood of recovering the original data sequence, weighted by the transition schedule:

$$\mathcal{L} = -\mathbb{E}_{t \sim \mathcal{U}(0,1), q(x_t \mid x)} \left[ \frac{\alpha_t'}{1 - \alpha_t} \log p_\theta(x_0 \mid x_t) \right], \tag{6}$$

$\alpha_t' = \alpha_t - \alpha_{t-1}$, i.e., the incremental change in retention probability with respect to timestep $t$. This objective ensures that the model learns to accurately approximate the reverse transition distribution across the entire transition trajectory.

## 2.2 DISCRETE DIFFUSION WITH MLLMS

A central design in our framework is the integration of a MLLM as the backbone to empower the discrete diffusion process to support unified medical multimodal generation. This selection is motivated by several key advantages of MLLMs: 1) MLLMs provide powerful cross-modal alignment priors from large-scale vision-language pre-training, which are crucial for ensuring the semantic and visual consistency of generated image-report pairs. 2) Their unified token-based representation offers inherent scalability, allowing the framework to accommodate diverse medical modalities—such as chest X-rays, CT scans, and pathology images—with minimal architectural changes. Our empirical results further validate this choice, demonstrating that MLLM backbones significantly outperform strong alternatives (e.g., DiT Peebles & Xie (2022))for medical report generation and image-report pair generation tasks (see Sec. D in Supplementary). Furthermore, our framework offers greater flexibility than prior MLLM-based diffusion models. While approaches like MMaDA (Yang et al., 2025) are limited to diffusion-specific backbones (Nie et al., 2025), our method is also compatible with a broader class of general-purpose auto-regressive (AR) MLLMs.

Since MLLMs are trained with causal (autoregressive) masking, while diffusion denoising relies on bidirectional context, MeDiM introduces two key adaptations: *causal mask removal* and *timestep embeddings*. In addition, we identify AdaLN as a critical normalization strategy. Together, these three components ensure a seamless integration of MLLMs into the discrete diffusion process, as detailed below.

**Causal Mask Removal.** Autoregressive MLLMs typically employ causal attention, which restricts each token to attend only to its previous context. While this constraint is suitable for left-to-right

multimodal tasks, it is insufficient for the discrete diffusion models, where medical image and report tokens must be mutually accessible to achieve cross-modal alignment. As shown in Fig. 7, causal attention leads to blurred boundaries in generated medical images and semantically inconsistent or disorganized content in generated reports. To address this issue, we remove the causal mask and enable bidirectional attention across the entire sequence to enable fully bidirectional information flow, which is essential for cross-modal consistency.

**Timestep Embeddings.** In discrete diffusion, the timestep determines the transition probabilities in the forward process (e.g., the probability of retaining a symbol or replacing it with the [MASK] token). The timestep provides the MLLM with essential transition-schedule information in the reverse diffusion. Without explicitly modeling temporal information, the MLLM backbone cannot recognize the current diffusion stage, which hinders its ability to apply appropriate discrete denoising and alignment strategies during the reverse process. Thus, we map each diffusion timestep into a continuous embedding vector and inject it into the MLLM backbone. The time embedding further modulates intermediate layers through AdaLN. This design ensures that the backbone is aware of the current stage in the reverse transition process.

**AdaLN Designs.** We further incorporate adaptive layer normalization (AdaLN) (Perez et al., 2018) and its variant AdaLN-Zero (Peebles & Xie, 2022) to enhance the stability and cross-modal consistency of our MLLM backbone. Unlike standard layer normalization with fixed affine transformations, AdaLN dynamically predicts the normalization parameters from timestep embeddings, ensuring that medical image tokens and report tokenizations are normalized under a shared yet context-aware transformation.

## 3 EXPERIMENTS

MeDiM can be applied to varied medical applications: medical image generation, medical report generation, and joint medical image–report pair generation. In the following sections, we evaluate MeDiM on these medical tasks and their impact to downstream tasks. We further discuss the impact of backbone choices in discrete diffusion models and validate the effectiveness of our architectural designs for medical multimodal generation in Sec. D.

### 3.1 DATASET

For training and evaluation, we adopt two widely used medical image–text datasets: the MIMIC-CXR (Johnson et al., 2019), a comprehensive chest X-ray with radiology reports, and the PathGen dataset (Sun et al., 2024b), a large-scale collection of pathology image–text pairs. Specifically, we use 368,960 chest X-ray pairs from MIMIC-CXR and 736,188 pathology pairs from PathGen, the pathology pairs are subsampled to balance the data distribution. For evaluation, we used 8,000 pathology pairs and adopt the MIMIC-CXR test set. Both datasets are employed for unified training or supervised fine-tuning (SFT) of MeDiM and baselines. This unified setting provides a comprehensive evaluation protocol in medical multimodal generation tasks.

### 3.2 SETTINGS AND METRICS

We adopt pretrained Liquid (Wu et al., 2024a) as the MLLM backbone of MeDiM, combined with the VQGAN (Esser et al., 2020) encoder from Chameleon (Team, 2024a) for image tokenization and the LLaMA tokenizer (Touvron et al., 2023) for text processing. The model is trained for 1M steps with a Warmup Cosine Annealing with Restarts learning rate schedule on 8 A100 GPUs for 160 GPU hours, starting from $1 \times 10^{-5}$. Training uses images at a resolution of $512 \times 512$ and text sequences truncated to a maximum of 256 tokens. During inference, MeDiM follows the MaskGIT inference strategy (Chang et al., 2022), progressively refining [MASK] tokens to generate coherent multimodal medical output. For medical image generation, we evaluate MeDiM using Fréchet inception distance (FID) and inception score (IS). For medical report generation, we adopt standard natural language generation metrics, including BLEU (B-1, B-2, B-3), METEOR (MTR), and ROUGE-L (R-L). For joint image–report pair generation, we assess cross-modal consistency using the Qwen2-VL (Team, 2024b) as an automatic evaluator, complemented by human evaluation.

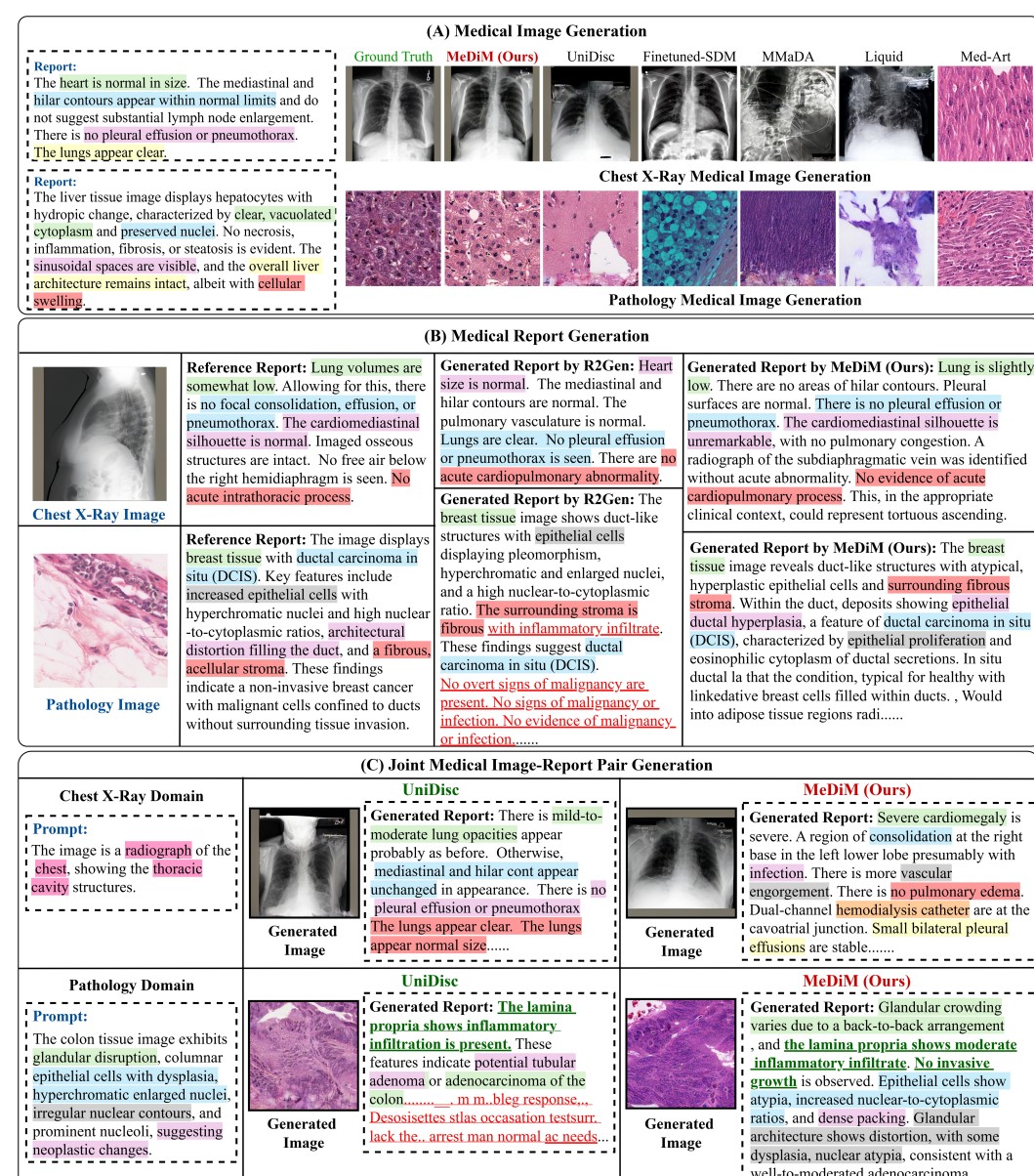

Figure 4: **Visual comparison of MeDiM against baselines on three tasks:** (A) medical image generation (unique colors indicate the alignment between the reference report and the images generated by MeDiM), (B) medical report generation (generated report and the reference are highlight with the same colors for matched content, while incorrect content is highlighted with red underlines), and (C) joint medical image–report pair generation (generated report and the prompt are highlight with the same colors for matched content, with green underlines denoting additional correct content consistent with the image, and red underlines marking incorrect content.).

And we evaluate downstream task performance to further verify the practical utility of the generated multimodal outputs in medical scenarios.

## 3.3 MEDICAL IMAGE GENERATION

For the medical image generation task, MeDiM retains all report tokenizations while replacing the image sequence with states. Thus, the generated images are conditioned on the corresponding reports. We compare MeDiM against three categories of baselines: (1) natural diffusion models (NA), such as the stable diffusion model (SDM) (Rombach et al., 2022); (2) specialized medical image gen-

Table 1: Quantitative comparison for chest X-ray image generation. * denotes fine-tuned models.

| Method | Type | FID ↓ | IS ↑ |
|---|---|---|---|
| SDM | NA | 120.28 | 2.92 |
| SDM (SFT) | NA | 78.97 | 2.91 |
| UniDisc | DDM | 82.54 | 2.82 |
| U-KAN | MED | 94.58 | 2.89 |
| Med-Art | MED | 168.92 | **3.82** |
| Liquid* | MLLM | 156.09 | 1.97 |
| MMaDA* | MLLM | 134.01 | 2.05 |
| MeDiM (Ours) | MLLM | **16.60** | 2.87 |

Table 2: Quantitative comparison for pathology image generation. * denotes fine-tuned models.

| Method | Type | FID ↓ | IS ↑ |
|---|---|---|---|
| SDM | NA | 159.93 | 2.59 |
| SDM (SFT) | NA | 55.76 | 4.03 |
| UniDisc | DDM | 80.99 | 4.15 |
| U-KAN | MED | 73.76 | 2.98 |
| Med-Art | MED | 107.45 | 2.93 |
| Liquid* | MLLM | 171.17 | 3.07 |
| MMaDA* | MLLM | 155.76 | 3.64 |
| MeDiM (Ours) | MLLM | **24.19** | **4.28** |

eration models (MED), including U-KAN (Li et al., 2025) and Med-Art (Guo et al., 2025a) (Notably, we introduce the class embedding of DiT into U-KAN, which enabled unified medical image generation.); (3) DDM: UniDisc (Swerdlow et al., 2025), and (4) multimodal generation–understanding models (MLLM), such as Liquid (Wu et al., 2024a) and MMaDA (Yang et al., 2025).

**Results.** Tab. 1 and Tab. 2 show the evaluation results of medical report–to–image generation across different medical benchmarks. In comparison to baselines, MeDiM demonstrates consistent SoTA performance. In Fig. 4, both MMaDA and Liquid, which were fine-tuned on medical image–report pairs, produce implausible distortions in the generated medical images, due to domain and task shift. The fine-tuned SDM fails to maintain high-fidelity results in medical image generation tasks and shows noticeable color shifts in pathology image synthesis. Although UniDisc can be applied to diverse medical image generation, the generated outputs are not always consistent with the corresponding reports (e.g., large blurred shadows in the lower lungs of chest X-rays while the reports describe "*The lungs appear clear*"). Compared to baselines, MeDiM generates medical images that exhibit higher fidelity and greater consistency with the medical reports.

Table 3: Comparison of our method (MeDiM) with different types of baselines on MIMIC-CXR and PathGen datasets. * indicates models that are fine-tuned under our dataset setting.

| Method | Type | MIMIC-CXR | | | | | PathGen | | | | |
|---|---|---|---|---|---|---|---|---|---|---|---|
| | | B-1 | B-2 | B-3 | MTR | R-L | B-1 | B-2 | B-3 | MTR | R-L |
| BLIP | NA | 0.240 | 0.125 | 0.053 | 0.125 | 0.265 | 0.106 | 0.054 | 0.031 | 0.140 | 0.236 |
| R2Gen | MED | 0.305 | 0.179 | 0.104 | 0.233 | **0.395** | 0.160 | 0.090 | **0.055** | 0.251 | **0.278** |
| R2GenCMN | MED | 0.266 | 0.132 | 0.061 | 0.223 | 0.225 | 0.142 | 0.069 | 0.037 | 0.248 | 0.267 |
| BLLM | MED | 0.252 | 0.152 | 0.070 | 0.201 | 0.220 | 0.113 | 0.053 | 0.018 | 0.154 | 0.229 |
| UniDisc | DDM | 0.270 | 0.137 | 0.075 | 0.224 | 0.206 | 0.109 | 0.039 | 0.012 | 0.180 | 0.113 |
| Liquid* | MLLM | 0.186 | 0.104 | 0.037 | 0.170 | 0.172 | 0.124 | 0.028 | 0.009 | 0.107 | 0.121 |
| MMaDA* | MLLM | 0.153 | 0.102 | 0.031 | 0.164 | 0.185 | 0.172 | **0.108** | 0.052 | 0.200 | 0.258 |
| MeDiM (Ours) | MLLM | **0.328** | **0.185** | **0.109** | **0.265** | 0.297 | **0.185** | 0.084 | 0.037 | **0.258** | 0.226 |

## 3.4 MEDICAL REPORT GENERATION

Diverse baselines are compared in the medical report generation task (NA), including BLIP (Li et al., 2022) as a representative of general-purpose captioning models, R2Gen (Chen et al., 2020), R2GenCMN (Chen et al., 2021), and BLLM (Liu et al., 2024a) as specialized medical report generation systems (MED), UniDisc (Swerdlow et al., 2025) in DDM, and Liquid (Wu et al., 2024a) and MMaDA (Yang et al., 2025) as multimodal generation–understanding models (MLLM). To further enhance the reliability of report evaluation in PathGen, we employ Qwen2-VL to filter the test set and select 5,000 high-quality medical reports, which are used as the ground truth (GT).

**Results.** Tab. 3 shows the performance of MeDiM on multiple benchmarks for medical image report generation. Compared with MLLM-based generative-understanding models such as Liquid, MMaDA, and UniDisc, as well as BLIP, a natural image captioning model, our approach achieves superior performance on both benchmarks. In comparison with specialized medical report genera-

tion approaches, such as R2Gen, R2GenCMN, and BLLM, our method also attains comparable or superior results. To further complement the quantitative evaluations, Fig. 4 presents qualitative comparisons of MeDiM and R2Gen. The R2Gen suffers from notable deficiencies, including semantic repetition, omission of salient details, and logical inconsistencies (see Fig. 4). In contrast, MeDiM reduces semantic redundancy and faithfully interprets clinically significant details.

### 3.5 JOINT MEDICAL IMAGE-REPORT PAIR GENERATION

In the joint medical image–report pair generation task, only UniDisc and MeDiM can simultaneously generate medical images and their corresponding reports. We adopt UniDisc as the baseline and evaluate the paired generation results in cross-modal consistency and downstream task performance.

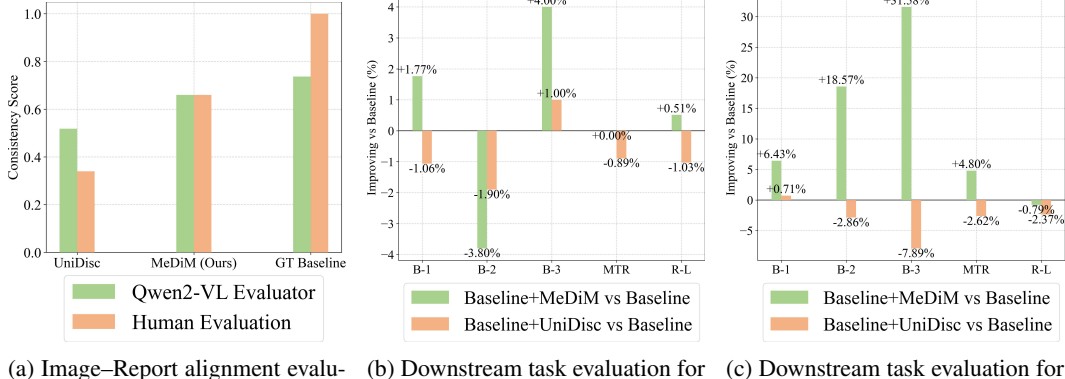

(a) Image–Report alignment evaluation.

(b) Downstream task evaluation for R2Gen in MIMIC-CXR.

(c) Downstream task evaluation for R2Gen in PathGen.

Figure 5: Quantitative evaluation of MeDiM on the joint medical image–report generation task.

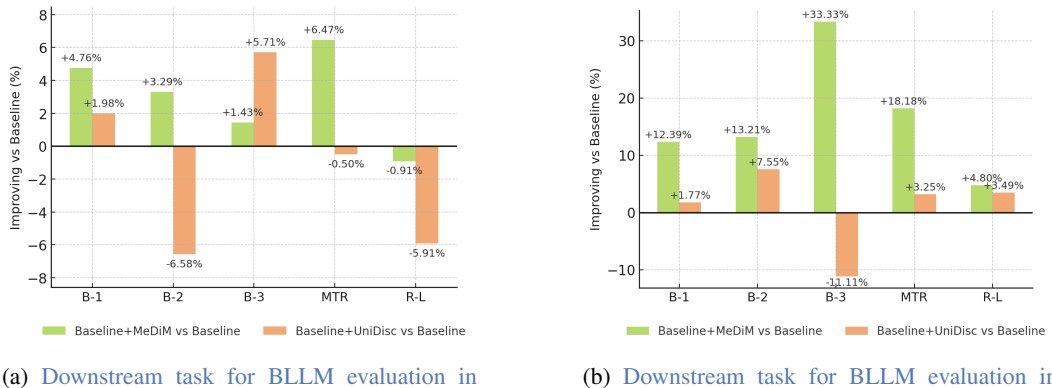

(a) Downstream task for BLLM evaluation in MIMIC-CXR.

(b) Downstream task for BLLM evaluation in PathGen.

Figure 6: Downstream evaluation with different medical VLM backbone.

**Alignment.** We evaluate the consistency of generated medical image–report pairs using the large VLM Qwen2-VL. We employ [MASK] sequences as inputs, unconditionally generate 8,000 pathology image–report pairs and 5,159 chest X-ray image–report pairs to support consistency evaluation. To further ensure reliability, we complement this with human evaluation, conducted on 100 unconditionally generated image–report pairs sampled in a 1:1 ratio between pathology and chest X-ray domains. As illustrated in Fig. 5a, large VLMs and human evaluators produce consistent judgments, both showing that MeDiM attains high confidence. Meanwhile, as observed in Fig. 4, MeDiM generates image–report pairs that adhere to the prompts and remain semantically consistent.

**Downstream Task.** In the downstream evaluation, we investigate the impact of synthetic medical image–report pairs on VLM performance under low-data medical settings. We first construct simple text prompts based on image modality, anatomical region, and pathology condition. These prompts are injected into the [MASK]-initialized sequences during inference, thereby guiding MeDiM or

UniDisc in conditionally generating 200k medical image–report pairs with a 1:1 balanced distribution. Then, we sample 200k real pairs from the MIMIC-CXR and PathGen training sets in a 1:1 ratio and merge them with the 200k synthetic pairs to form a balanced dataset. This dataset is used to train the medical report generation baseline R2Gen. As shown in Fig. 5b and Fig. 5c, the image–report pairs generated by MeDiM lead to noticeable gains in pathology visual analysis tasks. To further assess generalization, we additionally validate synthesis data on a recent medical VLM (BLLM (Liu et al., 2024a)) in Fig. 6a and Fig. 6b. Consistent with the results on R2Gen, MeDiM provides larger improvements than UniDisc, further reinforcing the robustness of MeDiM on the downstream VLM task.

## 4 CONCLUSION

This work presents MeDiM, a novel discrete diffusion model for medical multimodal generation. By learning the shared probabilistic distributions across modalities and domains, MeDiM acts as a flexible multimodal generator without modality-specific components. Furthermore, we improve the backbone with an MLLM equipped with distributional alignment priors based on empirical evidence. This achieves a unified representation space for both medical modalities. Extensive experiments across different medical modalities demonstrate robust performance for MeDiM in medical image generation, report generation, and joint image-text synthesis, highlighting the effectiveness of our MLLM backbone in the medical domain.

## 5 ETHICS STATEMENT

All authors of this work have read and commit to adhering to the ICLR Code of Ethics.

## 6 REPRODUCIBILITY

To ensure reproducibility, we provide full code in the *Supplementary Material*.

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

TECHNICAL APPENDICES

# A  RELATED WORK

**Medical Image Generation.** Owing to data scarcity, medical image generation is gaining increasing significance as a research focus. Early studies (Chartsias et al., 2017; Guo et al., 2020; Huo et al., 2018; Zhang et al., 2018) employed GANs to augment medical imaging data, improving the performance of downstream medical tasks. In recent years, diffusion models (Ho et al., 2020; Rombach et al., 2022) have shown stable training and promising performance in high-quality image generation tasks, motivating a growing body of work to explore their application in medical image generation (Lyu & Wang, 2022; Chambon et al., 2022; Güngör et al., 2023). Fast-DDPM (Jiang et al., 2025) and LLCM (Polamreddy et al., 2025) focus on accelerating the diffusion-based generation of medical images. Graikos et al. (2024) shows that self-supervised embeddings can replace fine-grained annotations by conditioning diffusion models on SSL features, enabling high-fidelity pathology and satellite image synthesis and large-scale cross-modal generation. EMIT-Diff (Zhang et al., 2024b) and MAISI (Guo et al., 2025b) incorporate ControlNet (Zhang et al.) to achieve anatomically or semantically controllable medical image synthesis. By introducing non-linear Kolmogorov-Arnold Networks (KANs) (Liu et al., 2024d) and globally modeling Transformers (Vaswani et al., 2017; Peebles & Xie, 2022) as diffusion backbones, U-KAN (Li et al., 2025) and Med-Art (Guo et al., 2025a) demonstrate state-of-the-art performance in generating medical images across diverse modalities, which motivates the consideration of backbone design for discrete diffusion models in MeDiM.

**Medical Report Generation.** Medical report generation is commonly facilitated by VLMs (Li et al., 2022; 2023; Zeng et al., 2023) endowed with domain-specific medical knowledge. Yang et al. (2022) enhanced the medical literacy of VLMs by combining general medical background with image-specific clinical knowledge, while Zhang et al. (2020) designed a medical knowledge graph to further enrich medical VLMs. R2Gen (Chen et al., 2020) and R2GenCMN (Chen et al., 2021) generate radiology reports based on memory-driven Transformer architectures. Chen et al. (2023) improved the interpretability of medical report generation by associating local medical image regions with medical terminology in the report, while Liu et al. (2024c) enhanced image–report alignment through fact-guided contrastive learning. RGRG (Tanida et al., 2023) employs region-level guidance for medical report generation, while Zhang et al. (2024a) addresses its limitation of neglecting shared attributes for each local region via attribute prototype guidance. Liu et al. (2024a) leveraged priors from LLMs to improve medical report generation. MLRG (Liu et al., 2025) enhances medical visual information by integrating multiple views of the same medical image. However, most existing approaches are limited to specific medical report generation tasks. To overcome this issue, we present MeDiM, which formulates the joint discrete distribution of medical images and reports independent of imaging modalities and organ-specific characteristics.

**Unified Multimodal Understanding and Generation.** Unified multimodal models aim to build an MLLM, modeling and reasoning over both image and textual sequences. Early studies (Ge et al., 2023; 2024; Sun et al., 2024a; 2023; Li et al., 2022) attempted to combine CLIP (Radford et al., 2021), which aligns the visual–language space, with LLMs to process images and text separately. Given that CLIP operates in a continuous visual space, some studies (Liu et al., 2024b; Team, 2024a; Wu et al., 2024b) have explored using VQ-VAE (Van Den Oord et al., 2017) to represent visual information as discrete sequences. Meanwhile, numerous studies (Yu et al., 2023; Lu et al., 2024; Xie et al., 2024a; Wu et al., 2025) discussed leveraging the strengths of both encoders. Studies such as LWM (Liu et al., 2024b) and Chameleon (Team, 2024a) have shown that discrete visual features can be integrated with language tokens into a unified sequence, facilitating joint cross-modal modeling. Although this design obviates the need for modality-specific components, it incurs substantial training costs. Liquid (Wu et al., 2024a) introduces a pre-trained LLM as the backbone to solve this issue. However, AR suffers from low inference efficiency. Given the efficiency in generation and success in understanding of discrete diffusion models, UniD3 (Hu et al., 2022) and UniDisc (Swerdlow et al., 2025) adopted them as a unified framework. MMaDA (Yang et al., 2025) further employs LLaDA (Nie et al., 2025) as the backbone for a unified discrete diffusion model. However, MMaDA is restricted to a single diffusion-based MLLM backbone, whereas MeDiM can leverage the wider spectrum of AR MLLM backbones, providing greater scalability.

Table 4: Ablation study for MeDiM on the backbone and components, evaluated with mean medical report understanding metrics (mB-1, mB-2, mB-3, mMTR, mR-L) and mean image generation metrics (mFID, mIS) on MIMIC-CXR and PathGen.

| Settings | Report Generation | | | | | Image Generation | |
|---|---|---|---|---|---|---|---|
| | mB-1 ↑ | mB-2 ↑ | mB-3 ↑ | mMTR ↑ | mR-L ↑ | mFID ↓ | mIS ↑ |
| MeDiM | 0.256 | 0.134 | 0.073 | 0.262 | 0.261 | 20.40 | 3.57 |
| w/ DiT backbone | 0.195 | 0.091 | 0.040 | 0.214 | 0.200 | 63.22 | 2.81 |
| w/ UniDisc backbone | 0.223 | 0.098 | 0.051 | 0.255 | 0.217 | 51.59 | 3.05 |
| w/o pretrained MLLM weight | 0.205 | 0.092 | 0.044 | 0.229 | 0.212 | 68.27 | 2.83 |
| w/o timestep embedding | 0.221 | 0.107 | 0.049 | 0.246 | 0.238 | 40.03 | 3.13 |
| w/o AdaLN designs | 0.232 | 0.108 | 0.056 | 0.247 | 0.249 | 32.68 | 3.16 |
| w/ causal mask | 0.152 | 0.068 | 0.025 | 0.142 | 0.179 | 143.72 | 2.02 |
| seed 1 | 20.37 | 3.50 | 0.252 | 0.128 | 0.075 | 0.260 | 0.265 |
| seed 2 | 20.55 | 3.36 | 0.260 | 0.131 | 0.068 | 0.274 | 0.258 |
| seed 3 | 0.256 | 0.134 | 0.073 | 0.262 | 0.261 | 20.40 | 3.57 |

Table 5: Out of distribution comparison in IU-Xray-RRG.

| Method | FID ↓ | IS ↑ |
|---|---|---|
| UniDisc | 173.59 | 2.33 |
| MMaDA | 205.17 | 1.85 |
| Liquid | 212.40 | 1.74 |
| **MeDiM (Ours)** | **129.42** | **2.60** |

Table 6: Comparison with medical expert generative models.

| Method | FID ↓ | IS ↑ |
|---|---|---|
| DiffInfinite | 143.05 | 2.38 |
| PathLDM | 176.25 | 2.29 |
| PixCell | 98.54 | 4.27 |
| **MeDiM (Ours)** | **24.19** | **4.28** |

# B  OUT OF DISTRIBUTION COMPARISON

To verify that MeDiM's improvements do not stem from dataset-specific fine-tuning, we evaluate the model on the unseen IU-Xray-RRG dataset, which differs significantly from MIMIC-CXR. As shown in Tab. 5, MeDiM achieves an FID of 129.42 and an IS of 2.60, outperforming UniDisc, MMaDA, and Liquid without any additional fine-tuning.

Although its performance is weaker than on MIMIC-CXR, this degradation is largely due to a structural distribution shift: MIMIC-CXR images contain consistent collimation borders, whereas IU-Xray-RRG does not. Overall, these results indicate that MeDiM maintains clear advantages under OOD conditions, and its gains arise from unified cross-modal modeling rather than overfitting to training datasets.

# C  COMPARISON WITH EXPERT MODEL

We compare MeDiM with several domain-specialized medical image generative models, including DiffInfinite (Aversa et al., 2023), PathLDM (Yellapragada et al., 2024), and PixCell (Yellapragada et al., 2025), which incorporate modality-specific inductive biases and customized training strategies for high-fidelity medical image synthesis. As shown in the Tab. 6, MeDiM achieves a substantially lower FID while maintaining a comparable or slightly higher IS. These results suggest that a unified discrete diffusion framework combined with an MLLM backbone can achieve a level of generative fidelity that is consistent with, and in some aspects approaches, that of specialized expert systems, even though it does not rely on architectures tailored to specific tasks or modalities.

We further evaluate MeDiM against LLavaRAD (Zambrano Chaves et al., 2025), an expert medical vision and language model designed for chest X-ray interpretation and clinical reporting. Across standard metrics for medical report generation in Tab. 7, MeDiM achieves higher scores. This indicates that a unified framework for multimodal generation

Table 7: Comparison with medical expert VLM.

| Method | B-1 | B-2 | B-3 | MTR | R-L |
|---|---|---|---|---|---|
| LLavaRAD | 0.144 | 0.075 | 0.038 | 0.154 | 0.261 |
| **MeDiM (Ours)** | **0.328** | **0.185** | **0.109** | **0.265** | **0.297** |

Figure 7: Visual results from the ablation study evaluating the effect of pre-trained MLLM weights.

and understanding provides a competitive and
complementary alternative to expert medical VLMs, demonstrating strong potential for broader
medical vision and language applications.

## D  ABLATION STUDY

We investigate the contribution of the MLLM backbone and its associated architectural components
to the overall performance of MeDiM. This analysis allows us to disentangle the impact of backbone
selection and design choices.

**Backbone.** We use DiT (Peebles & Xie, 2022) and a pretrained UniDisc backbone as backbone
baselines to examine the impact of backbone choice. We also assess the contribution of the pre-
trained distributional alignment prior in MLLM to MeDiM's performance. As shown in Tab. 4, re-
placing the MLLM backbone with different backbones or removing the pretrained MLLM weights
both lead to noticeable degradation for MeDiM's performance.

**Components.** We further conduct ablation studies on three architectural components of MeDiM to
assess their contributions: timestep embeddings, AdaLN designs, and causal mask removal. We re-
move the timestep embeddings and replace the AdaLN designs with a standard LayerNorm, thereby
eliminating temporal conditioning. For the AdaLN ablation, we retain timestep embeddings but
inject them directly into the token representations through weighted addition, without AdaLN com-
ponents. The ablation results in Tab. 4 demonstrate that timestep embeddings and AdaLN provide
essential temporal conditioning and effective feature modulation. In contrast, enforcing a causal
mask severely disrupts multimodal alignment, resulting in blurry image boundaries and semanti-
cally incoherent reports (see Fig. 7).

**Robust Test.** For the robustness analysis. we include results in Tab. 4 across multiple random seeds
to evaluate MeDiM's stability under variations in sampling randomness. The results show consis-
tently low variance in both image metrics and report-generation metrics, indicating that MeDiM
exhibits strong robustness and stable performance.

## E  LIMITATION

Although MeDiM achieves promising results on unified medical multimodal generation tasks, it has
not yet exceeded or matched expert medical models across all metrics at evaluation, with shortfalls
confined to a limited subset. We plan to improve MeDiM by incorporating MLLM backbones with
medical domain background knowledge, aiming to bridge the gap with SoTA medical expert models
and extending MeDiM to support a broader range of downstream medical tasks in the future.

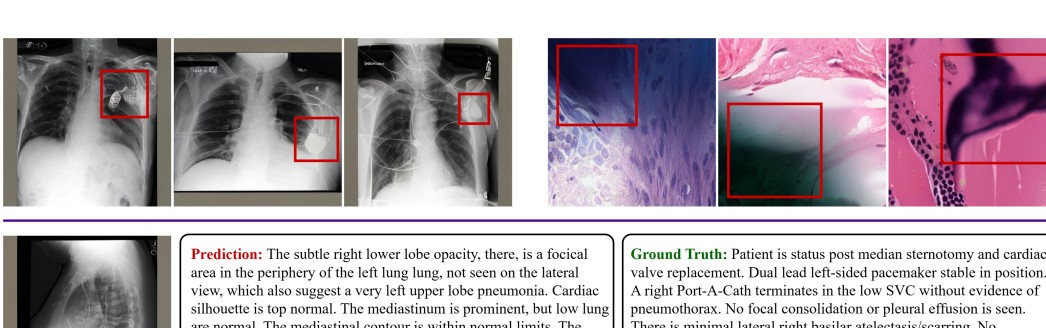

**Prediction:** The subtle right lower lobe opacity, there, is a focical area in the periphery of the left lung lung, not seen on the lateral view, which also suggest a very left upper lobe pneumonia. Cardiac silhouette is top normal. The mediastinum is prominent, but low lung are normal. The mediastinal contour is within normal limits. The pulmonary vasculature is not engorged. There is no pleural effusion, pulmonary edema, or pneumothorax.

**Ground Truth:** Patient is status post median sternotomy and cardiac valve replacement. Dual lead left-sided pacemaker stable in position. A right Port-A-Cath terminates in the low SVC without evidence of pneumothorax. No focal consolidation or pleural effusion is seen. There is minimal lateral right basilar atelectasis/scarring. No pulmonary edema is seen. The cardiac and mediastinal silhouettes are stable.

**Prediction:** The image exhibits acellular dense homogeneous areas. No signs of pathological changes like hyperctasis, mitotic figures, or cellular neoplasia are evident. This indicates a focal area possibly normal liver. Lackely visible parenchyma suggests a faint but well-defined region. The absence of cellular structures is not completely visualized on this image. The current image does not depict the usual exam.

**Ground Truth:** The tissue shows a homogenous pink matrix with scattered small, round to oval structures resembling follicular cysts, lacking cellular detail and possibly fluid-filled. No cellular atypia or malignancy is observed. These features suggest a benign ovarian cystic lesion, characterized by its non-malignant extracellular matrix and cyst-like structures.

Figure 8: Failure cases of MeDiM.

## F  FAILURE CASE

Although MeDiM performs robustly across both radiology and pathology domains, certain limitations can still be observed. Representative failure cases are presented in Fig. 8 to provide a transparent characterization of these behaviors.

(1) Local blurriness and mild color inconsistencies in pathology images. A small portion of synthesized pathology images exhibit slight blurriness or mild hue shifts. These artifacts mainly arise from the substantial color, stain, and texture variability across laboratories and slide preparation protocols. Rare staining styles and extreme color conditions can occasionally lead to unstable color reconstruction. Such cases are infrequent but are included here for completeness.

(2) Low-fidelity rendering of medical devices in chest X-rays. A few generated MIMIC-CXR samples contain imprecise or incomplete renderings of medical devices such as catheters, tubes, or monitoring lines (highlighted by red boxes in Fig. 8). This is primarily due to the long-tailed distribution of medical devices in real datasets, where rare or sparsely annotated devices provide limited visual evidence for reliable learning. Despite these inaccuracies, major anatomical structures remain well preserved.

(3) Semantic ambiguity when image cues are insufficient. In cases where the visual evidence is limited or structurally ambiguous—such as overlapping lateral shadows, faint pathological patterns, or unclear device boundaries—MeDiM may produce semantically ambiguous predictions. Examples include confusing devices with lesions or generating descriptions that do not precisely correspond to the underlying organ shown in the image. Such cases highlight that, similar to clinical practice, ambiguous visual cues can challenge even strong multimodal models.

Together, these examples provide insight into current limitations and offer guidance for future improvements in color stabilization, rare object modeling, and uncertainty-aware reasoning.

## G  DECLARATION OF LLM TOOL USAGE

During the preparation of this manuscript, I used OpenAI's GPT-4.1 model for minor language refinement and smoothing of the writing. The LLM tool was not used for generating original content, conducting data analysis, or formulating core scientific ideas. All conceptual development, experimentation, and interpretation were conducted independently without reliance on LLM tools. The other points involving the use of LLMs have already been highlighted in the paper.

