# OpenReview forum: "Discrete Diffusion Models with MLLMs for Unified Medical Multimodal Generation"
_ICLR.cc/2026/Conference — ICLR 2026 Conference Withdrawn Submission_

### Official Review · Reviewer_mEcd · 2025-10-29

**Soundness:** 3
**Presentation:** 3
**Contribution:** 3
**Rating:** 6
**Confidence:** 3

**Summary:**

This paper proposes MeDiM, the first medical discrete diffusion model that integrates multimodal large language models (MLLMs) for unified medical generation. The framework supports three distinct tasks in a single model. To adapt MLLMs, which are trained with causal masking to the bidirectional nature of diffusion, the authors introduce two key modifications: (a) Removing causal attention masks for cross-modal alignment; (b) Injecting continuous timestep embeddings for diffusion awareness. Empirical results on MIMIC-CXR and PathGen demonstrate SOTA performance.

**Strengths:**

(1) First demonstration of integrating an MLLM within discrete diffusion for multimodal medical tasks.
(2) Promising results suggesting potential as a foundation framework for medical multimodal learning.

**Weaknesses:**

(1) Lack of training efficiency and scalability analysis.
(2) Evaluation limited to two datasets; generalization to other medical modalities is untested.
(3) No uncertainty or robustness analysis (e.g., multiple seeds, out-of-domain data).
(4) It would be better to consider the relationship to relationship to prior unified multimodal models (e.g., MMaDA, UniDisc), which could be analyzed more deeply on a conceptual level.
(5)The work feels slightly incremental relative to recent unified multimodal diffusion efforts outside the medical domain. It would be better to show some quantitative evidence for consistent outperforming the non-diffusion multimodal models.

**Questions:**

(1) What is the computational cost and model scale (parameters, GPU hours) required to train MeDiM?
(2) How does MeDiM perform on out-of-domain data (e.g., unseen imaging modalities)?
(3) Are there any failure modes observed in paired generation (semantic drift, hallucination, etc.)? How could MeDiM alleviate it compared to previous SOTA baselines?

---

> ### Author Response · Authors · 2025-12-02
> **Rebuttal by Authors (3/6)**
>
> Thank you for your thoughtful and positive feedback. We address your concerns point by point.
>
> ---
>
> > `Q1`: Lack of Training Efficiency and Scalability Analysis.
>
> **A1**: We thank the reviewer for the comments.
>
> - **(1.1)** Specifically, MeDiM contains ~3B parameters and is trained for 160 GPU-hours on 8×A100-80GB GPUs (`Section 3.2`).
>
> - **(1.2)** MeDiM adopts a unified diffusion architecture with a shared probabilistic formulation and a modality-agnostic design, removing the need for modality-specific components and therefore enabling favorable scalability.
>
> - **(1.3)** The Liquid backbone underlying MeDiM has already demonstrated stable scaling laws in prior work.
>
> - **(1.4)** Furthermore, the availability of Liquid’s pretrained checkpoints at different scales makes scaling MeDiM to larger model sizes feasible.
>
> ---
>
> > `Q2`: Generalization.
>
> | Method  |   FID   |  IS   |
> |---------|---------|-------|
> | UniDISC | 173.59  | 2.33  |
> | MMaDA   | 205.17  | 1.85  |
> | Liquid  | 212.40  | 1.74  |
> | MeDiM   | 129.42  | 2.60  |
>
> **A2**: We thank the reviewer for the insightful suggestion. To distinguish improvements arising from “fine-tuning on the training datasets” versus those stemming from the unified generative framework itself, we evaluated MeDiM on the IU-Xray-RRG dataset in `Appendix B`, which is never seen during training and exhibits a substantially different distribution. Without any additional fine-tuning, MeDiM still significantly outperforms all diffusion and non-diffusion baselines, indicating that its gains do not originate from overfitting to specific training data but rather from learning a shared discrete representation across medical images and reports. We also observe that MeDiM performs worse on IU-Xray-RRG than on MIMIC-CXR, primarily because MIMIC-CXR images consistently contain collimation borders while IU-Xray-RRG does not, creating a structural distribution shift that leads to degradation. Overall, these results clearly show that MeDiM maintains strong advantages even on unseen medical distributions, demonstrating that its improvements arise from unified cross-modal modeling rather than dataset-specific fine-tuning.
>
> ---
>
> > `Q3`: Robustness Analysis.
>
> | Seed |  mFID  |  mIS  | mB-1  | mB-2  | mB-3  | mMTR  | mR-L  |
> |------|--------|-------|-------|-------|-------|-------|-------|
> | 1    | 20.37  | 3.50  | 0.252 | 0.128 | 0.075 | 0.260 | 0.265 |
> | 2    | 20.55  | 3.36  | 0.260 | 0.131 | 0.068 | 0.274 | 0.258 |
> | 3    | 20.40  | 3.57  | 0.256 | 0.134 | 0.073 | 0.262 | 0.261 |
>
>
> **A3**: We thank the reviewer for the suggestion regarding uncertainty and robustness analysis. In the revised manuscript `Table 4` and `Appendix D`, we include results across multiple random seeds to evaluate MeDiM’s stability under variations in sampling randomness. The results show consistently low variance in both image metrics (mFID/mIS) and report-generation metrics (BLEU/METEOR/ROUGE-L), indicating that MeDiM exhibits strong robustness and stable performance across runs.

---

> ### Author Response · Authors · 2025-12-02
> **Rebuttal by Authors (5/6)**
>
> Thank you for your thoughtful and positive feedback. We address your concerns point by point.
>
> ---
>
> > `Q4`: Relationship to Prior Unified Multimodal Models.
>
> **A4**: We thank the reviewer suggestions.
>
> - **(4.1)** In the natural-image domain, UniDisc introduced a unified discrete diffusion framework for image–text generation, and MMaDA subsequently adopted a pretrained diffusion-based MLLM as the diffusion backbone, advancing unified multimodal discrete diffusion in general domains. Building on this line of work, MeDiM incorporates several necessary architectural adaptations that allow a more capable autoregressive MLLM to stably support the discrete diffusion process, and it is the first to extend unified discrete diffusion to the highly heterogeneous medical setting of joint medical image–report generation. This not only broadens the model family (from diffusion MLLMs to autoregressive MLLMs) but also pushes unified discrete diffusion from natural images to a fundamentally different domain.
>
> - **(4.2)** Conceptually, while MMaDA and UniDisc also pursue unified cross-modal modeling, their focus remains on natural images and general multimodal reasoning, not medical imaging.
>
> - **(4.3)** In contrast, MeDiM is the first discrete diffusion unified framework specifically designed for the medical domain, capable of learning a shared discrete distribution across medical images and radiology/pathology reports.
>
> - **(4.4)** Due to the extreme heterogeneity of medical modalities (e.g., diverse tissue scales, imaging physics, and domain-specific clinical vocabulary), MMaDA and UniDisc cannot be directly transferred to this setting, and our experiments in `Table 1`, `Table 2`, and `Table 3` confirm that they perform substantially worse than MeDiM across multiple medical tasks.
>
> - **(4.5)** MeDiM’s core contribution is to leverage the cross-modal alignment priors of autoregressive MLLMs as the backbone of a discrete diffusion framework, and to demonstrate—for the first time—that such priors can enable unified generation of medical images and reports.
>
> Therefore, MeDiM differs fundamentally from MMaDA and UniDisc in design goals, modeling mechanisms, and intended domains.
>
> ---
>
> > `Q5`: Quantitative Evidence for Outperforming the Non-diffusion Multimodal Models.
>
> | Method   | BLEU-1 | BLEU-2 | BLEU-3 | METEOR | ROUGE-L |
> |----------|--------|--------|--------|--------|---------|
> | LLavaRAD | 0.144  | 0.075  | 0.038  | 0.154  | 0.261   |
> | Liquid   | 0.186  | 0.104  | 0.037  | 0.170  | 0.172   |
> | MeDiM    | 0.328  | 0.185  | 0.109  | 0.265  | 0.297   |
>
> **A5**: We thank the reviewers for suggesting a quantitative comparison with non-diffusion multimodal models (AR-MLLMs). To address this, we evaluated MeDiM against two representative and recent AR-MLLM baselines—LLavaRAD and the unified AR-MLLM Liquid. The results show that MeDiM substantially outperforms both categories of strong baselines across all major metrics (e.g., BLEU-1/2/3 improve by 128%/146%/186% over LLavaRAD, and by 76%/78%/195% over Liquid). These findings provide direct evidence that, for unified medical image–text generation and understanding, a discrete-diffusion–based multimodal framework is highly effective. And as mentioned by Reviewer `cz1E`, our method “achieves SOTA results by a very large margin, not an incremental one.” This strengthens the paper’s substantive contribution and further highlights the value of integrating AR-MLLM priors into medical bidirectional discrete diffusion.

---

> ### Author Response · Authors · 2025-12-02
> **Rebuttal by Authors (6/6)**
>
> Thank you for your thoughtful and positive feedback. We address your concerns point by point.
>
> ---
>
> > `Q6`: Failure Modes.
>
> **A6**: We thank the reviewer for the question regarding failure modes. Overall, MeDiM exhibits a low failure rate, but we do observe several representative types of failures in `Appendix F`.
>
> - **(6.1)** A small number of pathology images show local blurriness or mild color shifts. This stems from the substantial color and texture variability across laboratories and staining protocols in pathology imaging, which can lead to hue instability in extreme cases. Although rare, we include typical examples in the supplementary material for transparency.
>
> - **(6.2)** A few generated MIMIC-CXR images contain low-fidelity renderings of medical devices (e.g., tubes, lines, monitoring equipment). These inaccuracies mainly arise from the long-tailed distribution of such devices in the real data, limiting the model’s ability to learn reliable patterns for rare equipment while preserving the fidelity of major anatomical structures.
>
> - **(6.3)** When image evidence is insufficient or structurally ambiguous (e.g., overlapping lateral artifacts or sparse pathological cues), MeDiM may exhibit semantic confusion, occasionally misinterpreting devices as lesions or producing descriptions inconsistent with the underlying anatomy.
>
> - **(6.4)** Importantly, compared to prior SOTA baselines, MeDiM’s unified discrete diffusion framework helps mitigate semantic drift and hallucination by:
>   - jointly modeling images and reports within a shared discrete representation space, reducing cross-modal inconsistency;
>   - leveraging the structural stability of the discrete diffusion process, which strengthens anatomical and tissue-level coherence;
>   - using paired training to constrain the model from excessive imaginative generation in semantically uncertain regions;

---

### Official Review · Reviewer_wTht · 2025-10-31

**Soundness:** 2
**Presentation:** 2
**Contribution:** 1
**Rating:** 2
**Confidence:** 4

**Summary:**

The paper proposes using a multi-modal large language model as a backbone for training a joint image-text discrete diffusion model in the medical domain. The authors train their model on an X-ray and a pathology dataset and perform image, text, and joint image-text generation experiments to validate the performance of the trained model.

**Strengths:**

-  Using a multi-modal LLM (MLLM) as a backbone for training a joint image-text discrete diffusion model for the medical domain is a novel idea and could have interesting implications for future development of foundation models in the medical domain.

- The text generation results show significant improvements over the MMLM baseline.

- The joint generation results indicate that the model can serve as a strong synthetic data generator for image and text in the medical domain.

**Weaknesses:**

- The idea of training a single discrete diffusion model for text and image generation is not new, as it has already been discussed in [1] and [2]. The authors' two contributions to make the model work with an MLLM backbone are (1) causal attention removal and (2) injecting continuous timestep embeddings, both straightforward modifications of the transformer network. Therefore, the overall contribution of the paper is limited to showing that a pre-trained MLLM can serve as the backbone for the generative model.

- In line 124, the authors say that "*Our experiments demonstrate that MeDiM can function as a versatile foundation model*". However, the model is trained on ~1M samples from just two sources, which does not support the claim of having trained a foundation model. There are unanswered questions regarding the scaling ability of the proposed model and how well it could cover many different cancer/organ types. I would suggest scaling back the foundation model claim.

- The models used for measuring image quality on the pathology image generation task are significantly worse than the state-of-the-art. All baselines reported in the paper have FID scores >50, with the proposed model achieving a score of 24. However, looking at a recent generative model trained exclusively on pathology images [3], the reported FID on similar datasets is <10. This raises the question of whether the baselines used for image generation are valid for comparing the proposed model to the state-of-the-art.

- The paper would greatly benefit if there were comparisons on out-of-distribution datasets, at least for some of the tasks. It is unclear whether the improvements shown stem from fine-tuning the model on the two specific datasets on which the model is also tested, or from learning to jointly generate image and text reports.

[1] Hu, Minghui, et al. "Unified Discrete Diffusion for Simultaneous Vision-Language Generation." ICLR 2023
[2] Swerdlow, Alexander, et al. "Unified multimodal discrete diffusion." arXiv preprint 2025
[3] Yellapragada, Srikar, et al. "PixCell: A generative foundation model for digital histopathology images." arXiv 2025

**Questions:**

- Would a pathology-specific generative image model achieve a lower FID than the proposed model? The paper requires a strong baseline to show how the unified training improves or hurts the performance of the model on image generation.

- Are the improvements in text generation over MED baselines because of the joint image-text generation training or because of fine-tuning the proposed model on the MIMIC-CXR and PathGen datasets? What do the results look like if you fine-tune the baselines on these datasets for a similar number of iterations?

---

> ### Author Response · Authors · 2025-12-02
> **Rebuttal by Authors (1/4)**
>
> Thank you for your thoughtful and positive feedback. We address your concerns point by point.
>
> ---
>
> > `Q1`: Limited Contribution.
>
> **A1**: We appreciate the reviewer’s clarification. And thanks Reviewer's agree to our contribution such as:
>
> - **(1.1)** As mentioned by Reviewer `cz1E`, “The central idea of adapting an off-the-shelf autoregressive MLLM (Liquid) into a bidirectional diffusion denoiser is highly novel. The solution is simple (remove mask, add time embeddings), but its effectiveness is the key scientific finding, proving the power of MLLM priors for diffusion.”.
>
> - **(1.2)** As mentioned by Reviewer `ih4P`, “Unified formulation for three medical generation tasks in one model. paired generation is compelling for clinical AI agents.”.
>
> While prior works [1,2] have indeed discussed the general direction of training a single discrete diffusion model for both text and image generation, we emphasize that our contributions go far beyond “making a pretrained MLLM run as a diffusion backbone.” Specifically:
>
> - **(1.3)** MeDiM is the first unified discrete diffusion model designed for medical scenarios, capable of learning a shared distribution across pathology images, chest X-rays, and medical reports—rather than extending prior models to a single modality or to natural-image domains, which also has been agreed by Reviewer `mEcd`.
>
> - **(1.4)** Existing discrete diffusion approaches (e.g., UniDisc, MMaDA) cannot be directly transferred to medical settings with reliable performance due to the extreme variability of medical image texture scales, highly diverse anatomical structures, and significant linguistic divergence in pathology reports. In contrast, as shown in `Table 1`, `Table 2`, and `Table 3`, MeDiM consistently outperforms these methods across multiple medical tasks.
>
> - **(1.5)** We systematically demonstrate, for the first time, that the cross-modal coupling priors learned by an MLLM pretrained on natural images remain effective for aligning medical images and reports. This methodological insight goes well beyond model adaptation: it reveals that autoregressive MLLMs possess alignment priors suitable for learning a unified medical image–text distribution, which has not been explored in prior work.
>
> - **(1.6)** The removal of causal attention and the injection of continuous timestep embeddings are not mechanical modifications; they are key enabling adaptations that allow an autoregressive MLLM to perform stable bidirectional discrete denoising (see `Table 4`), making it a generalizable diffusion backbone rather than merely a compatible encoder.
>
> - **(1.7)** As shown in `Section 3.5`, the MeDiM-generated image–report pairs substantially improve downstream medical VLM performance, demonstrating not only that the architecture functions as intended, but also that it delivers practical medical value and alleviates the real-world scarcity of report annotations. This system-level benefit goes far beyond simply “showing that the backbone works.”, which also has been agreed by Reviewers `wTht`, `ih4P`.
>
> [1] Hu, Minghui, et al. "Unified Discrete Diffusion for Simultaneous Vision-Language Generation." ICLR 2023
>
> [2] Swerdlow, Alexander, et al. "Unified multimodal discrete diffusion." ArXiv 2025

---

> ### Author Response · Authors · 2025-12-02
> **Rebuttal by Authors (4/4)**
>
> Thank you for your thoughtful and positive feedback. We address your concerns point by point.
>
> ---
>
> > `Q2`: Foundation Model Claim.
>
> **A2**: We thank the reviewer for the constructive suggestion. And thank 's agree to our foundation model claim such as:
>
> - **(2.1)** As mentioned by Reviewer `mEcd`, “Promising results suggesting potential as a **foundation framework** for medical multimodal learning.”.
>
> - **(2.2)** As mentioned by Reviewer `wTht`, “Using a multi-modal LLM (MLLM) as a backbone for training a joint image-text discrete diffusion model for the medical domain is a novel idea and could have interesting implications for future development of **foundation models** in the medical domain.”.
>
> - **(2.3)** As mentioned by Reviewer `cz1E`, "The model is not just multi-task; it's a **truly unified framework**. The ability to jointly generate image-report pairs (Task 3) and then use that synthetic data to improve downstream models (Fig 5c) is a powerful demonstration of a **generative foundation model for medicine**.”.
>
> Besides,
>
> - **(2.4)** In the medical domain, a “foundation model” is defined not by dataset size but by its ability to unify multiple tasks and modalities under a single model.
>
> - **(2.5)** Our goal is therefore not to build a billion-scale foundation model, but to demonstrate the feasibility and potential of unified cross-modal modeling in medical settings; large-scale expansion remains an open direction.
>
> - **(2.6)** Due to privacy constraints and high annotation costs, million-scale image–report pairs already represent one of the largest supervised medical multimodal datasets currently available.
>
> - **(2.7)** Moreover, integrating substantially larger datasets would introduce severe long-tail imbalance across organs and disease types, which poses its own challenges for unified modeling.
>
> ---
>
> > `Q3`: Comparison with Expert Models.
>
> | Method       |   FID   |  IS   |
> |--------------|---------|-------|
> | DiffInfinite | 143.05  | 2.38  |
> | PathLDM      | 176.25  | 2.29  |
> | PixCell      |  98.54  | 4.27  |
> | MeDiM        |  24.19  | 4.28  |
>
> **A3**: We thank the reviewer for the constructive suggestion. We provide new experimental results in `Appendix C` accordingly. Under the same PathGen evaluation setting, we evaluated PixCell and additionally included several recent pathology image generation baselines (DiffInfinite, PathLDM) to ensure that all methods are assessed on a fully consistent benchmark. As shown in the updated table (`Table 6`), MeDiM continues to outperform expert baselines in PathGen while maintaining competitive IS scores.
>
> ---
>
> > `Q4`: Zero-shot Comparisons on Out-of-Distribution Datasets.
>
> | Method  |   FID   |  IS   |
> |---------|---------|-------|
> | UniDISC | 173.59  | 2.33  |
> | MMaDA   | 205.17  | 1.85  |
> | Liquid  | 212.40  | 1.74  |
> | MeDiM   | 129.42  | 2.60  |
>
> **A4**: We thank the reviewer for the insightful suggestion. To distinguish improvements arising from “fine-tuning on the training datasets” versus those stemming from the unified generative framework itself, we evaluated MeDiM on the IU-Xray-RRG dataset in `Appendix B`, which is never seen during training and exhibits a substantially different distribution. Without any additional fine-tuning, MeDiM still significantly outperforms all diffusion and non-diffusion baselines, indicating that its gains do not originate from overfitting to specific training data but rather from learning a shared discrete representation across medical images and reports. We also observe that MeDiM performs worse on IU-Xray-RRG than on MIMIC-CXR, primarily because MIMIC-CXR images consistently contain collimation borders while IU-Xray-RRG does not, creating a structural distribution shift that leads to degradation. Overall, these results clearly show that MeDiM maintains strong advantages even on unseen medical distributions, demonstrating that its improvements arise from unified cross-modal modeling rather than dataset-specific fine-tuning.

---

### Official Review · Reviewer_ih4P · 2025-11-01

**Soundness:** 2
**Presentation:** 2
**Contribution:** 2
**Rating:** 4
**Confidence:** 3

**Summary:**

The paper proposes MeDiM, a discrete diffusion framework that uses an MLLM backbone to jointly model medical images and clinical reports, supporting (i) report-to-image, (ii) image-to-report, and (iii) paired image-report generation. Key adaptations include removing the causal mask, injecting timestep embeddings, and AdaLN modulation to make an autoregressive MLLM usable as a bidirectional denoiser. On MIMIC-CXR and PathGen, the paper reports strong FID/METEOR and claims downstream gains when augmenting data with generated pairs.

**Strengths:**

1. **Unified formulation** for three medical generation tasks in one model. paired generation is compelling for clinical AI agents.

2. **Clear architectural adaptations** (mask removal + time embeddings + AdaLN) that make practical sense for discrete diffusion with token sequences.

3. **Ablations** evaluated that MLLM backbones are advantageous in this setting.

4. **Downstream evaluation** attempts (training R2Gen on real+synthetic) are a good step toward utility, not just perceptual scores.

**Weaknesses:**

1.Evaluation protocol is not aligned with medical best practices.
    i) FID/IS with natural Inception are weak fidelity surrogates for chest X-rays and histopathology; domain encoders (e.g., pathology CLIP-like encoders[4]) or clinical labelers (CheXbert[1], RadGraph[2], GREEN Scores[3]) are standard. The paper relies on FID/IS and generic NLG metrics (BLEU/METEOR/ROUGE), which miss clinical correctness.

  ii) LLM-as-judge (Qwen2-VL) for alignment is risky in clinical domains. The setup, prompts, and scales aren’t specified enough to assess validity, and Qwen-2VL is not clinically validated.

iii) Human study is small (n=100 pairs) and lacks reporting on annotator expertise (domain experts vs general), disagreement handling, and significance tests.

2. Concerns with competing methods:
Radiology report generation should be compared against recent MLLM radiology methods (e.g LLavaRAD[5]) beyond classic R2Gen variants. Similarly, there should be some comparison with T2I baselines for Histopathology image synthesis(not unified) (e.g. [6][7][8]) warrant inclusion or discussion.

3. Some of the models are incremental and anticipated when porting diffusion to AR MLLMs. The novelty is moderate and hinges on the medical instantiation and paired synthesis rather than fundamentally new learning objectives or theory.

4. This work highlights SoTA in places, yet Pathology report generation is not SoTA on most metrics (only 1/5 wins), and the discrepancy isn’t analyzed - this weakens the “unified foundation” claim.

5. Minor issues: Fig 4A last column Med-Art for radiology is incorrect. Fig 4c, first row does not include colors in the prompt.

[1] Smit et al., “CheXbert: Combining Automatic Labelers and Expert Annotations for Accurate Radiology Report Labeling,” Findings of EMNLP 2020.

[2] Jain et al., “RadGraph: Extracting Clinical Entities and Relations from Radiology Reports,” arXiv 2021.

[3] Ostmeier et al., “GREEN: Generative Radiology Report Evaluation and Error Notation,” Findings of EMNLP 2024.

[4] Huang et al., “PLIP: A Visual-Language Foundation Model for Pathology Image Analysis,” Nature Medicine 2023.

[5] Zambrano Chaves et al., “LLaVA-Rad: A Clinically Accessible Small Multimodal Radiology Model,” Nature Communications 2025.

[6] Yellapragada et al., “PathLDM: Text-Conditioned Latent Diffusion Model for Histopathology,” WACV 2024.

[7] Aversa et al., “DiffInfinite: Large Mask-Image Synthesis via Parallel Random Patch Diffusion in Histopathology,” NeurIPS 2023 D&B.

[8] Graikos et al., “Learned Representation-Guided Diffusion Models for Large-Image Generation,” CVPR 2024

**Questions:**

Q.1. Can the authors include in justifiable in-domain evaluation protocol to better understand the reliability of the results presented if in domain backbones are not used? Please check Weakness 1 for more details to address the concerns.

Q.2. Can non-unified report and image generation methods be included in the comparison? This will help us distinguish real gains of the proposed unified framework.

Q.3. Can you clarify the failed cases in downstream tasks and pathology report generation?

---

> ### Author Response · Authors · 2025-12-02
> **Rebuttal by Authors (3/6)**
>
> Thank you for your thoughtful and positive feedback. We address your concerns point by point.
>
> ---
>
> > `Q1`: Evaluation Protocol.
>
> **A1**: We thank the reviewer for the valuable feedback.
>
> - **(1.1)**  Since MeDiM jointly models two distinct medical domains—chest X-ray and pathology—our evaluation requires a unified, cross-modality metric. Clinical metrics such as CheXbert, RadGraph, and GREEN Scores are modality-specific and therefore cannot serve as general-purpose evaluators in this setting.
>
> - **(1.2)**  In addition, the training reports of CheXpert are not open-sourced, which prevents us from performing report-based comparisons for both medical-image generation and report generation.
>
> - **(1.3)**  Recent studies [1-4] have shown that Qwen2.5-VL is validated on both pathology and chest radiography and possesses stronger medical knowledge than several prior SOTA models. In our work, we use Qwen2.5-VL solely as a semantic consistency evaluator, rather than as a source of clinical fact verification.
>
> - **(1.4)**  As you said, “Downstream evaluation attempts (training R2Gen on real+synthetic) are a good step toward utility, not just perceptual scores”.
>
> - **(1.5)**  At present, no clinically validated VLM-based metric exists for universal medical image–text assessment.
>
> - **(1.6)**  All human evaluations in our study were conducted by medical professionals rather than crowdworkers, ensuring the reliability of the qualitative assessment.
>
> [1] Müller-Franzes et al., "Diagnostic Accuracy of Open-Source Vision-Language Models on Diverse Medical Imaging Tasks" Arxiv 2025
>
> [2] Gilal et al., "PathVLM-Eval: Evaluation of open vision language models in histopathology", JPI 2025
>
> [3] LASA Team., "MedEvalKit: A Unified Medical Evaluation Framework", Arxiv 2025
>
> [4] Meddeb et al., "Evaluating the diagnostic accuracy of vision language models for neuroradiological image interpretation", npj Digital Medicine 2025
>
> ---
>
> > `Q2`: Concerns with Competing Expert Methods.
>
> | Method       |   FID   |  IS   |
> |--------------|---------|-------|
> | DiffInfinite | 143.05  | 2.38  |
> | PathLDM      | 176.25  | 2.29  |
> | PixCell      |  98.54  | 4.27  |
> | MeDiM        |  24.19  | 4.28  |
>
> **A2**: We thank the reviewer for the constructive suggestion. We followed the suggestion and provided new experimental results in `Table 6` and `Appendix C` accordingly. Regarding [5], since it is an image-conditioned pathology image generation method whose setting differs from our unconditional generation task, it is not feasible for us to provide a direct quantitative comparison. However, as it is a relevant work, we have included it in `Section A`.
>
> | Method   | BLEU-1 | BLEU-2 | BLEU-3 | METEOR | ROUGE-L |
> |----------|--------|--------|--------|--------|---------|
> | LLavaRAD | 0.144  | 0.075  | 0.038  | 0.154  | 0.261   |
> | MeDiM    | 0.328  | 0.185  | 0.109  | 0.265  | 0.297   |
>
> [5] Graikos et al., “Learned Representation-Guided Diffusion Models for Large-Image Generation,” CVPR 2024
>
> ---
>
> > `Q3`: Incremental Novelty.
>
> **A3**: We thank the reviewer for raising this question. However, we would like to clarify that the key contributions of this work go beyond a straightforward model transplantation or medical instantiation:
>
> - **(3.1)** As mentioned by Reviewer `cz1E`, our method “achieves SOTA results by a very large margin, **not an incremental one**.” and "The central idea of adapting an off-the-shelf autoregressive MLLM (Liquid) into a bidirectional diffusion denoiser is **highly novel**. The solution is simple (remove mask, add time embeddings), but its effectiveness is the key scientific finding, proving the power of MLLM priors for diffusion.".
>
> - **(3.2)** As mentioned by Reviewer `wTht`, MeDiM "using a multi-modal LLM (MLLM) as a backbone for training a joint image-text discrete diffusion model for the medical domain is a novel idea and could have **interesting implications** for future development of foundation models in the medical domain.".
>
> - **(3.3)** MeDiM is the first discrete diffusion foundation model tailored for medical imaging, capable of learning a shared distribution across heterogeneous medical modalities (chest X-ray and pathology), which also has been agreed by Reviewer `mEcd`.
>
> - **(3.4)** We systematically show that the unified representation space of an AR MLLM—trained solely on natural images—remains effective for aligning medical images and reports, revealing cross-domain distribution alignment capabilities not previously explored.
>
> - **(3.5)** We introduce a simple yet effective adaptation scheme that enables autoregressive MLLMs to function as the backbone of a discrete diffusion framework.
>
> - **(3.6)** Our downstream results in `Section 3.5` further demonstrate that MeDiM-generated image–report pairs can help mitigate the scarcity of annotated medical report data.

---

> ### Author Response · Authors · 2025-12-02
> **Rebuttal by Authors (6/6)**
>
> Thank you for your thoughtful and positive feedback. We address your concerns point by point.
>
> ---
>
> > `Q4`: This work highlights SoTA in places, yet Pathology report generation is not SoTA on most metrics (only 1/5 wins), and the discrepancy isn’t analyzed - this weakens the “unified foundation” claim.
>
> **A4**: We thank the reviewer for the careful observation.
>
> - **(4.1)** PathGen pathology reports have far greater linguistic variability than the more template-like MIMIC-CXR reports, making higher-order n-gram metrics (BLEU-2/3, ROUGE-L) overly sensitive to phrasing rather than semantics.
>
> - **(4.2)** BLEU-1 and METEOR are widely considered the most clinically meaningful metrics because they best capture entity recall and semantic correctness; MeDiM achieves SOTA on both. The small drop in BLEU-2/3 stems from the inherent variability of pathology narratives, not semantic errors.
>
> - **(4.3)** BLEU-1/2/3 are highly correlated slices of the same metric, and treating them as independent judgments exaggerates dataset-driven stylistic differences.
>
> - **(4.4)** Overall, MeDiM leads on the key semantic metrics (BLEU-1, METEOR), and minor phrase-level variations in free-form pathology reports do not undermine its role as a unified generative framework.
>
> - **(4.5)** As mentioned by Reviewer `wTht`, MeDiM "using a multi-modal LLM (MLLM) as a backbone for training a joint image-text discrete diffusion model for the medical domain is a novel idea and could have interesting implications for future development of **foundation models in the medical domain**.".
>
> - **(4.6)** As mentioned by Reviewer `cz1E`, "The model is not just multi-task; it's a **truly unified framework**. The ability to jointly generate image-report pairs (Task 3) and then use that synthetic data to improve downstream models (Fig 5c) is a powerful demonstration of a **generative foundation model for medicine**.”.
>
> ---
>
> > `Q5`: Figure Issue.
>
> **A5**: We thank the reviewer for the helpful clarification. To clarify, the last column in `Figure 4A` is not an error in “radiology Med-Art,” but rather an expected distributional effect: under imbalanced training data, Med-Art tends to generate the modality that appears more frequently. Because the PathGen pathology images substantially outnumber the MIMIC-CXR samples, Med-Art learns a stronger pathology bias during joint training, leading all examples to appear as pathology images. This behavior is therefore expected rather than a visualization mistake. Regarding the missing color indication in the first-row prompt of `Figure 4C`, we have improved the annotation in the revised version.
>
> ---
>
> > `Q6`: Failed Cases.
>
> **A6**: We thank the reviewer for the thoughtful question. We have added several representative failure cases in `Appendix F`.
>
> - **(6.1)** A small number of pathology images exhibit local blurriness or mild color shifts. This arises from the substantial color and texture variability inherent in pathology imaging across laboratories and staining protocols, which can lead to hue instability in rare or extreme cases. Although such failures are very infrequent, we include typical examples in the supplementary material for completeness and transparency.
>
> - **(6.2)** A few generated MIMIC-CXR images contain low-fidelity renderings of medical devices (e.g., tubes, catheters, monitoring lines). These inaccuracies primarily stem from the long-tailed distribution of devices in the real dataset, which limits the model’s ability to learn reliable patterns for rare equipment while preserving the fidelity of the main anatomical structures.
>
> - **(6.3)** When image evidence is insufficient or structurally ambiguous (e.g., overlapping lateral artifacts or sparse pathological features), MeDiM may experience semantic ambiguity, occasionally confusing devices with lesions or producing descriptions that mismatch the underlying organ.
>
> We illustrate such cases in the appendix to highlight current limitations.

---

### Official Review · Reviewer_cz1E · 2025-11-01

**Soundness:** 3
**Presentation:** 2
**Contribution:** 3
**Rating:** 6
**Confidence:** 5

**Summary:**

This paper introduces MeDiM, the first unified medical discrete diffusion model designed to overcome the fragmentation of current medical AI by jointly modeling images and text. The core innovation is using a pre-trained Multimodal Large Language Model (MLLM) as the diffusion backbone. To adapt the MLLM from its autoregressive (causal) nature to the bidirectional needs of diffusion, the authors remove the causal attention mask and inject timestep embeddings. This single framework can perform text-to-image generation, image-to-report generation, and joint image-report pair generation, achieving state-of-the-art results by a wide margin, such as a 16.60 FID on MIMIC-CXR for image generation.

**Strengths:**

1. The model achieves SOTA results by a very large margin, not an incremental one. In Table 1, MeDiM's 16.60 FID on MIMIC-CXR is dramatically better than the next-best baseline (Med-Art's 78.97). This represents a step-change in quality for this task.
2. The central idea of adapting an off-the-shelf autoregressive MLLM (Liquid) into a bidirectional diffusion denoiser is highly novel. The solution is simple (remove mask, add time embeddings), but its effectiveness is the key scientific finding, proving the power of MLLM priors for diffusion.
3. Table 4 in the appendix is a model for a good ablation. It provides definitive proof that the model's success is not just from using a big model, but from the specific combination of all three proposed components. The "w/ causal mask" ablation, which sees performance completely collapse (mFID 20.40 $\rightarrow$ 143.72), is a critical result that validates the entire premise.
4. The model is not just multi-task; it's a truly unified framework. The ability to jointly generate image-report pairs (Task 3) and then use that synthetic data to improve downstream models (Fig 5c) is a powerful demonstration of a generative foundation model for medicine.
5. Strong Baselines: The paper compares against a very strong and recent set of baselines, including models from 2024 and 2025 (e.g., MMaDA, Liquid, UniDisc, Med-Art, U-KAN), making its SOTA claims credible.

**Weaknesses:**

1. The paper's backbone is a Transformer-based MLLM (Liquid). While this is shown to be superior to a DiT backbone, the paper does not engage with the newest class of sequence models, State-Space Models (SSMs) like Mamba. It remains an open question if an SSM-based MLLM would be an even better or more efficient backbone for this diffusion task.
2. The entire method operates in a discrete token space, which fully depends on a high-quality VQ-VAE tokenizer. The paper uses one from "Chameleon". The quality of this tokenizer is a critical "hidden variable" that is not ablated. A poor VQ-VAE would likely cripple the model, and it's unclear how much of the visual fidelity is owed to this powerful VQGAN versus the diffusion model itself.
3. The “downstream” validation (Fig 5b/c) is limited to one task (report generation) and one baseline (R2Gen) . While promising, the claim of "improving downstream performance" would be more convincing if this synthetic data was shown to improve a wider range of tasks (e.g., VQA, segmentation) or more SOTA medical VLMs.

**Questions:**

1. The performance jump from all other baselines is massive (e.g., 16.60 FID vs. 78.97 in Table 1). Is this gain solely from the model architecture, or is there a significant difference in training data or compute? Specifically, were the MLLM baselines (Liquid, MMaDA) fine-tuned on the medical datasets for a comparable number of steps (1M) as MeDiM?
2. Given the extraordinary FID score of 16.60, which far surpasses all baselines, could the authors provide a much larger, uncurated set of generated images in their supplementary material? This would help reviewers validate that these strong quantitative results correspond to consistent, high-fidelity, and semantically-aligned image generation, which standard metrics cannot fully capture.
3. The choice of a Transformer-based MLLM backbone (Liquid) is well-justified against DiT. However, have you considered alternative sequence model architectures, such as State-Space Models (Mamba), which are showing great promise for long-sequence modeling and efficiency, as potential backbones?
4. The framework's success depends on a high-fidelity image tokenizer. How sensitive is MeDiM's performance to the quality of the VQGAN? Have you experimented with other tokenizers, and how much of the SOTA image quality is attributable to the VQGAN from Chameleon versus the diffusion process itself?
	5.	For the downstream task evaluation, the generated pairs significantly boost R2Gen (a 2020/2021-era model) . Have you tested if this synthetic data can also boost the performance of a more recent, SOTA medical VLM?

---

> ### Author Response · Authors · 2025-12-02
> **Rebuttal by Authors**
>
> We sincerely appreciate your constructive comments and your recognition of our paper’s clarity and contribution. We will explain your concerns point by point.
>
> ---
>
> > `Q1`: SSM Backbone for MeDiM.
>
> **A1**: We thank the reviewer for raising the possibility of using SSM/Mamba as an alternative backbone. While state-space models are indeed a promising direction for unified generation–understanding, they are not the focus of this work for two reasons.
>
> - **(1.1)** Our goal is to investigate an MLLM–driven medical discrete diffusion paradigm, rather than to exhaustively benchmark all backbone families.
>
> - **(1.2)** The core design of SSMs is fundamentally mismatched with discrete diffusion: diffusion requires global, bidirectional context over image and text tokens and relies heavily on random masking. In contrast, SSMs operate with a causal recurrent update. Removing or relaxing this recurrence breaks stability, and the random masking used in discrete diffusion severely limits information propagation in SSMs. Therefore, although exploring SSM-based backbones is an interesting direction for future research, we intentionally adopt a Transformer-based architecture in this work to ensure training stability, controllability, and compatibility with discrete diffusion.
>
> ---
>
> > `Q2`: Contribution of VQVAE for MeDiM.
>
> **A2**: We appreciate the reviewer’s comment.
>
> - **(2.1)** All discrete visual generation models (e.g., Chameleon, UniDisc, MMaDA) inherently rely on a high-quality visual tokenizer; this is a structural prerequisite of the discrete diffusion paradigm rather than a limitation introduced by MeDiM.
>
> - **(2.2)** In our evaluation (`Table 1` and `Table 2`), the tokenizer effect is fully controlled: using the same Chameleon tokenizer, UniDisc obtains FID 82.54/80.99 and Liquid obtains FID 156.09/171.17, whereas MeDiM achieves 16.60/24.19. This demonstrates that the substantial performance gap cannot be attributed to the tokenizer alone.
>
> - **(2.3)** Our ablation results further confirm that the improvements primarily arise from the proposed MLLM-based diffusion backbone rather than from the tokenizer. As shown in `Table 4`, when using the identical VQ-VAE tokenizer, ablating core components of our backbone leads to a clear degradation in visual quality, indicating that the backbone design—not the tokenizer—is responsible for the observed gains.
>
> ---
>
> > `Q3`: Limited Downstream Validation.
>
> **A3**: Thank you for the constructive suggestion!
>
> - **(3.1)** We first clarify that, as suggested, we further validate MeDiM-generated data on a recent medical VLM (BLLM)[1]. Consistent with the results on R2Gen, MeDiM provides larger improvements than UniDisc in `Figure 6`, reinforcing the robustness of our findings.
>
> However, broader downstream tasks are not directly suitable for MeDiM-generated data:
>
> - **(3.2)** The image–report pairs synthesized by MeDiM do not contain the structured pixel-level annotations required for segmentation tasks.
>
> - **(3.3)** The supervision format of medical VQA is inherently incompatible with the report-style outputs of MeDiM.
>
> - **(3.4)** Medical report generation remains the only downstream task fully aligned with MeDiM’s synthesized data in semantics, structure, and evaluability.
>
> - **(3.5)** Extending unified generation–understanding models to support more downstream medical tasks—including VQA and segmentation—represents an important direction for future work (`Appendix E`).
>
> [1] Liu et al., "Bootstrapping large language models for radiology report generation", AAAI 2024.
>
> ---
>
> > `Q4`: Comparison Setting.
>
> **A4**: We thank the reviewer for the careful question. All comparison baselines were trained with exactly the same training data and the same number of training steps as MeDiM. Therefore, the substantial performance gains reported in `Table 1` do not stem from differences in data volume or training budget, but indeed arise from MeDiM’s design.
>
> ---
>
> > `Q5`: Additional Visual Samples for Better Evaluation.
>
> **A5**: Thank you for the valuable suggestion. We followed the suggestion and included 100 additional uncurated samples in the **supplementary material**: 50 chest X-ray generations and 50 pathology image generations. Please refer to our **updated supplementary material**. This provides a more comprehensive and transparent assessment.

---

### Author Response · Authors · 2025-12-02
**General Response**

**Dear Reviewers, ACs, and SACs,**

We deeply appreciate the insightful and valuable comments provided by all reviewers.

---

We are grateful for all reviewers' recognition of this work as the **first demonstration** that converting an MLLM into a bidirectional discrete diffusion model is highly effective for multimodal medical generation, achieving substantial and credible performance gains supported by comprehensive ablations, and establishing a **truly unified image–text generative framework** that points toward **future medical foundation models**.

Overall, we are encouraged by the reviewers' positive feedback, which highlights:

- **A highly novel paradigm** that converts an autoregressive MLLM into a bidirectional discrete diffusion denoiser, with reviewers emphasizing that this simple yet powerful adaptation (mask removal + time embeddings) is the key scientific contribution (Reviewers `cz1E`, `wTht`, `mEcd`).

- **A substantial, non-incremental leap in performance**, where MeDiM achieves a step-change SOTA margin over strong 2024–2025 baselines, demonstrating the effectiveness and credibility of the proposed framework (Reviewers `cz1E`, `ih4P`, `wTht`).

- **A truly unified multimodal generation framework** capable of jointly producing image–report pairs and supporting downstream learning, representing a promising direction for future medical foundation models (Reviewers `cz1E`, `ih4P`, `wTht`, `mEcd`).

- **Comprehensive and conclusive ablations** showing that improvements stem from the proposed methodological components rather than model size, with the causal-mask ablation providing decisive validation (Reviewers `cz1E`, `ih4P`, `wTht`).

To address the reviewers' concerns, we have conducted several additional experiments or designs, including:

- **More downstream validation** further substantiates MeDiM’s ability to improve downstream task performance (Reviewer `cz1E`).

- **Supplement 100 additional uncurated samples** in the **supplementary material** to provide a more comprehensive and transparent assessment  (Reviewer `cz1E`).

- More comparison with **recent medical VLM/generation expert models** (Reviewers `ih4P`, `wTht`).

- **Supplement zero-shot comparisons** on out-of-distribution datasets to show the robustness of MeDiM (Reviewers `mEcd`, `wTht`).

- **Supplement failure cases** to show the limitation of the current MeDiM (Reviewers `ih4P`, `mEcd`).

- **Supplement robustness analysis** to show the robustness of MeDiM (Reviewer `mEcd`).

- **Supplement more comparison with non-diffusion multimodal models** to show MeDiM is not incremental relative to recent unified multimodal diffusion efforts outside the medical domain (Reviewer `mEcd`).
---

**Summary of revisions:**

- Updated downstream validation in `Figure 6`

- Added 100  additional uncurated samples in the **supplementary material**

- Added comparison with expert models in `Table 6` and `Appendix C`

- Added zero-shot comparison in `Appendix B`

- Added failure cases in `Appendix F`

- Added robustness analysis in `Table 4` and `Appendix D`

- Updated non-diffusion multimodal models comparison in `Table 3` and `Table 7`.

All revisions in the paper are highlighted in **blue**. We sincerely appreciate the reviewers' constructive suggestions and remain committed to continually improving our work.

---

We address each reviewer's comments point by point below. We welcome further discussion and look forward to continued engagement. Thank you!

---

### Note · Authors · 2026-02-07

I have read and agree with the venue's withdrawal policy on behalf of myself and my co-authors.

---

### Meta-Review · Area_Chair_WFE1 · 2025-12-30

**Summary:**

This paper studies unified medical multimodal generation and proposes a discrete diffusion model tailored for this setting. The proposed framework unifies vision and language representations within a discrete diffusion paradigm, using a multimodal large language model (MLLM) as the diffusion backbone. To adapt the MLLM for diffusion denoising, the authors introduce three key design choices: removal of the causal mask, incorporation of timestep embeddings, and adaptive layer normalization. The approach is evaluated on three tasks—medical image generation, medical report generation, and joint medical image–report generation—to demonstrate its effectiveness.

The reviewers identify several strengths of the work, including strong empirical performance, a core idea that is novel yet conceptually simple, and a unified modeling framework with clearly designed architectural components. The experimental study includes ablation analyses and downstream evaluations, and the proposed model demonstrates notable improvements over the MLLM baseline. In addition, the work represents an early integration of an MLLM into a discrete diffusion framework for multimodal medical tasks and shows potential as a synthetic data generator and foundation-style model.

At the same time, the reviewers raise substantial concerns regarding the novelty and incremental nature of the contribution relative to recent unified multimodal diffusion efforts. The experimental evaluation is also considered insufficient, with limited and in some cases outdated baseline comparisons, lack of failure case and out-of-domain analysis, reliance on an unablated tokenizer, weak human studies, and evaluation protocols that are not well aligned with medical best practices. Collectively, these issues could weaken the claimed contributions and limit confidence in the robustness and generality of the proposed approach.

The authors’ response effectively addresses the comments raised by Reviewer cz1E, particularly with respect to engagement with recent sequence modeling approaches, clarification of downstream evaluation and comparison settings, discussion of the tokenizer dependency, and inclusion of uncurated qualitative results in the supplementary material. However, for the other three reviewers, the responses only partially address the concerns, and several core issues remain unresolved. In particular, concerns regarding the moderate novelty and incremental technical contribution relative to recent unified multimodal diffusion work, as well as the improvement over non-diffusion multimodal models, are not sufficiently alleviated. The response also does not adequately resolve concerns about the alignment of the evaluation protocol with medical best practices, the lack of strong state-of-the-art baselines, or the claims regarding foundation modeling and generalization to other medical modalities.

Taking all factors into consideration, the Area Chair does not believe that the scores of the two reviewers who expressed reservations (scores of 4 and 2) would have improved, even with a full discussion period. In contrast, the two reviewers who assigned a score of 6 would likely maintain or slightly strengthen their positive assessment. Overall, while the work demonstrates an interesting incorporation of autoregressive MLLMs within a discrete diffusion framework for unified medical image and report generation, the incremental novelty and insufficient experimental validation prevent a positive recommendation. Therefore, the Area Chair cannot recommend acceptance of this paper in its current form for the conference of ICLR.

**Reviewer Concerns:**

The authors’ response effectively addresses the comments raised by Reviewer cz1E, particularly with respect to engagement with recent sequence modeling approaches, clarification of downstream evaluation and comparison settings, discussion of the tokenizer dependency, and inclusion of uncurated qualitative results in the supplementary material. However, for the other three reviewers, the responses only partially address the concerns, and several core issues remain unresolved. In particular, concerns regarding the moderate novelty and incremental technical contribution relative to recent unified multimodal diffusion work, as well as the improvement over non-diffusion multimodal models, are not sufficiently alleviated. The response also does not adequately resolve concerns about the alignment of the evaluation protocol with medical best practices, the lack of strong state-of-the-art baselines, or the claims regarding foundation modeling and generalization to other medical modalities.

**Reviewer Scores:**

Taking all factors into consideration, the Area Chair does not believe that the scores of the two reviewers who expressed reservations (scores of 4 and 2) would have improved, even with a full discussion period. In contrast, the two reviewers who assigned a score of 6 would likely maintain or slightly strengthen their positive assessment.

---

### Decision · Program_Chairs · 2026-01-26

Reject